# Hybrid endosomal coats contain different classes of sorting nexins

Navin Gopaldass 🆔 ✉, Sudeshna Roy Chowdhury 🆔, Ana Catarina Alves 🆔, Lydie Michaillat Mayer 🆔,
Véronique Comte-Misérez & Andreas Mayer 🆔 ✉

## Abstract

**Endosomes are protein sorting stations, where multiple membrane coats form tubulovesicular carriers exporting proteins to the Golgi, the plasma membrane, or endo-lysosomal compartments. Distinct classes of sorting nexins are assumed to form distinct homogeneous coats that define the endosomal sorting routes and their cargos. Snx3 and the SNX-BAR proteins Vps5-Vps17 belong to different sorting-nexin classes. They can form homogeneous retromer-dependent coats that differ in structure and in their modes of membrane association and cargo recognition. Here, we describe the formation of hybrid coats between purified SNX-BARs, Snx3, and their cargos. Hybrid coats assemble at variable subunit ratios and diameters and show greater membrane-scaffolding activity than homogeneous coats. In vivo, Snx3 and SNX-BARs co-localise and mutually impact the sorting of their respective cargos. Although simultaneous binding of Snx3- and SNX-BARs to Retromer is sterically prohibited, hybrid coats incorporate both SNXs in a common complex, probably linked by retromer oligomerisation. We hence propose that SNX-BARs and Snx3 form retromer-mediated hybrid coats in novel, stoichiometrically adaptable configurations that allow the adjustment of endosomal carriers for transporting varying ratios of cargo.**

**Keywords** Retromer; Endosomes; Lysosomes; Membrane Traffic; Yeast
**Subject Category** Membranes & Trafficking

## Introduction

Retromer is a key player in endosomal recycling. This protein complex was discovered in yeast as a pentamer, consisting of two subcomplexes: a heterodimer of the SNX-BAR sorting nexins Vps5 and Vps17, and a heterotrimer of Vps26, Vps29 and Vps35 (Seaman et al, 1998). Today, the term Retromer is used to refer to this heterotrimer alone. Retromer is recruited to endosomal membranes through SNX proteins. In yeast, Retromer binds the SNX-BAR dimer Vps5-Vps17 or Snx3 (Seaman et al, 1998; Horazdovsky et al, 1997; Harrison et al, 2014; Strochlic et al,

2007). In mammalian cells, Retromer binds SNX3 (Harterink et al, 2011; Vardarajan et al, 2012; Chen et al, 2013), but it interacts with the homologues of the yeast Vps5/Vps17 dimer only indirectly. These homologues, the SNX-BARs SNX1/SNX2 and SNX5/SNX6, form the ESCPE complex (Simonetti et al, 2017, 2019; Lopez-Robles et al, 2023). Unlike the yeast SNX-BARs, ESCPE-1 interacts with Retromer through an additional sorting nexin, SNX27 (Lopez-Robles et al, 2023; Chandra et al, 2025, 2022; Guo et al, 2024; Simonetti et al, 2022).

CryoET structures of Retromer bound to either SNX-BARs or Snx3 have been solved (Leneva et al, 2021; Kovtun et al, 2018; Chen et al, 2025; Kendall et al, 2022, 2020). These structures revealed that Retromer can oligomerize to form arch-like structures that cross-link SNX proteins. They show Retromer associating with SNX3 and SNX-BARs in different ways and cross-linking them in different patterns (Fig. 1C). For example, while Vps26 contacts the membrane in Snx3/Retromer coats, this is not the case in the SNX-BAR/Retromer coats, where Vps26 has no contact with lipids and binds the membrane only indirectly through the SNX-BAR layer. Both the SNX-BAR and the SNX3-based Retromer coats show an SNX layer that covers the membrane extensively but with limited regularity (Leneva et al, 2021; Kovtun et al, 2018). The binding sites for Snx3 and the SNX-BARs are positioned such that Retromer could only bind one or the other type of sorting nexin, but not both at the same time.

In current models, SNXs usually define distinct recycling pathways and contribute to the selection of specific sets of cargo proteins (Buser and Spang, 2023; Weeratunga et al, 2020; van Weering et al, 2010). For example, yeast Vps10 is mainly ascribed to the Retromer/Vps5-Vps17 complex, while Ear1, Ste13, Kex2 or Ftr1 are addressed by the Retromer/Snx3 complex (Strochlic et al, 2008, 2007; Voos and Stevens, 1998; Harrison et al, 2014; Suzuki et al, 2019). Similarly, in mammals, SNX27- and SNX3-Retromer complexes recycle specific cargo to the plasma membrane and the TGN, respectively (Temkin et al, 2011; Lauffer et al, 2010; Yang et al, 2018; Ghai et al, 2011; Lee et al, 2016; Steinberg et al, 2013; Harterink et al, 2011; Wassmer et al, 2009; Cullen and Korswagen, 2012; McGough et al, 2014, 2018; Tian et al, 2021; Cui et al, 2018). Sorting occurs at the endosome surface, where coat components can be enriched in domains, which generate tubulovesicular cargo carriers (Varandas et al, 2016; Thompson et al, 2007; Derivery et al, 2012, 2009; Antón-Plágaro et al, 2024; Puthenveedu et al, 2010). However, there is a priori no reason why different coat complexes

Department of Immunobiology, University of Lausanne, Epalinges, Switzerland. ✉E-mail: navin.gopaldass@unil.ch; andreas.mayer@unil.ch

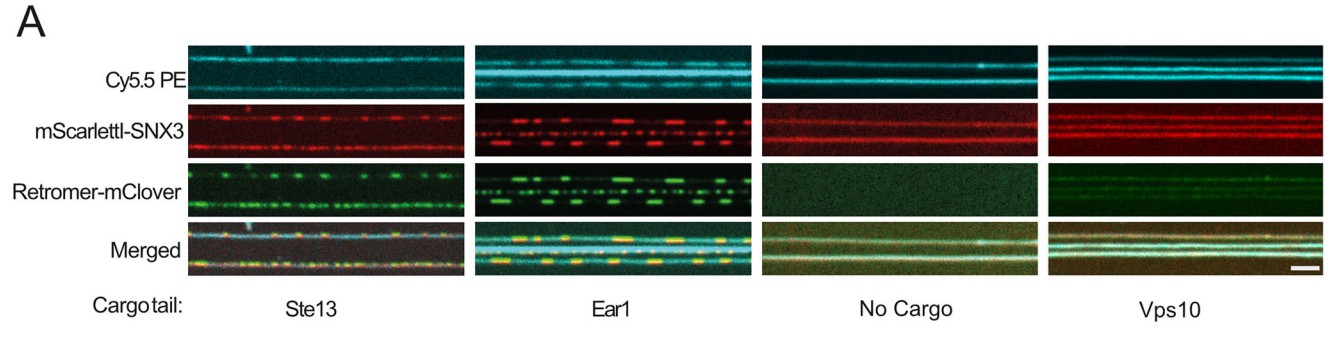

**A**

Cargo tail: Ste13    Ear1    No Cargo    Vps10

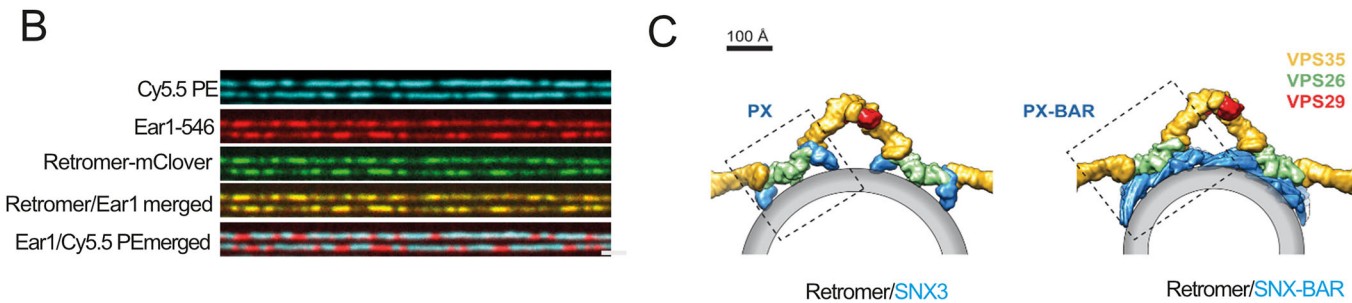

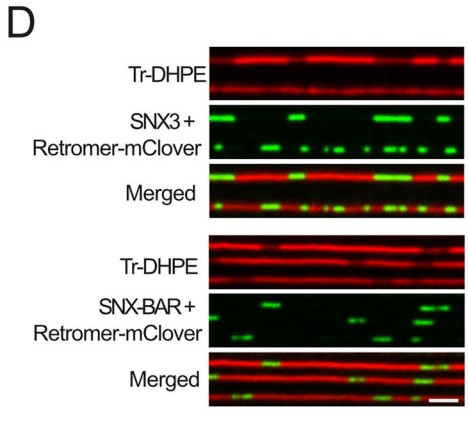

**B**

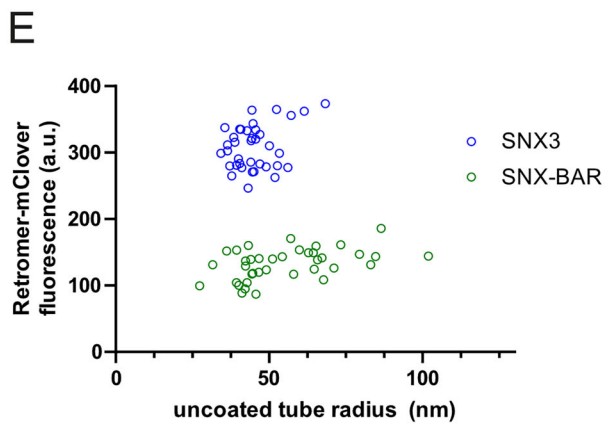

**C**

100 Å

PX                           PX-BAR

VPS35
VPS26
VPS29

Retromer/SNX3              Retromer/SNX-BAR

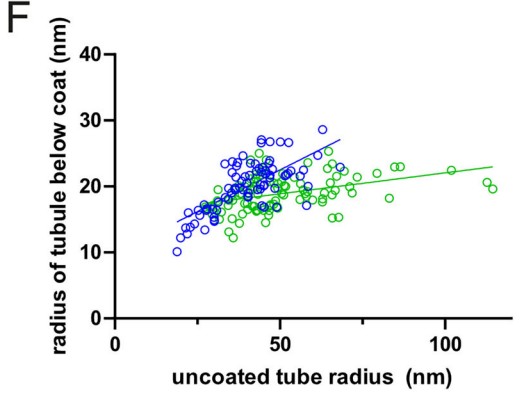

**D**

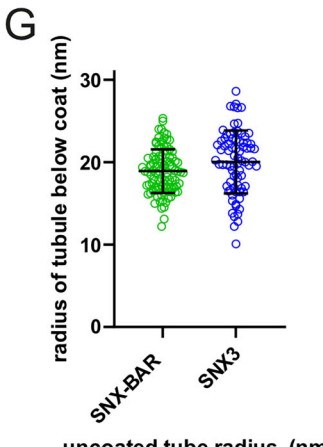

**E**

○ SNX3
○ SNX-BAR

**F**

**G**

◄ **Figure 1. Formation of Snx3-Retromer coats and comparison with SNX-BAR-Retromer coats.**

(A) Snx3-Retromer coat formation. SMTubes were formed in the presence of the lipid marker Cy5.5-PE and incubated with 10 μM cargo peptide for 10 min. Then, 50 nM of ^mScarletI^Snx3, Retromer^mClover (carrying mClover on Vps29) and 10 μM unlabelled cargo peptide were added. Tubes were imaged by confocal microscopy after 5 min of incubation. (B) Cargo concentration by Snx3-Retromer coats. Tubes were formed as in (A), pre-incubated with fluorescently labelled cargo peptide Ear1-547 for 5 min, and further incubated with 50 nM Snx3, 50 nM Retromer^mClover and 10 μM Ear1-547 until coat formation (2–3 min). After a brief wash with phosphate-buffered saline (PBS) buffer, tubes were imaged by spinning disc confocal microscopy. (C) Differing modes of Retromer membrane interaction in the Snx3 and SNX-BAR coats, taken from (Leneva et al, 2021). (D) Comparison of Retromer coats based on Snx3 and SNX-BARs. SMTubes, which were labelled through Texas Red DHPE (TR-DHPE), were incubated for 5 min with 25 nM SNX-BAR/Retromer^mClover or 50 nM Snx3/Retromer^mClover and imaged as in (B). (E) Quantification of the density of Retromer^mClover on Snx3/Retromer or SNX-BAR/Retromer coats as a function of the uncoated tube radius. The uncoated tube radius was quantified through the fluorescence intensity of TR-DHPE. The fluorescence had been calibrated by comparison to SMTs constricted by Dynamin I, which constricts lipid tubules to a defined radius of 11.2 nm (Roux et al, 2010). (F) The lipid tube radius under Snx3/Retromer or SNX-BAR/Retromer coats was quantified using the same approach as in (E). (G) Pooled data from (F), not considering the uncoated tube radii. $n = 38$. All scale bars are 2 μm. Bars represent the means and the standard deviation. Source data are available online for this figure.

might not be recruited simultaneously into the nascent endosomal carriers, since many of them are quite compatible with the highly curved surface of these tubulovesicular structures. This might lead to the formation of heterogeneous coats, where a single tubule contains a series of homogeneous subdomains with different SNX/Retromer coats, or to "hybrid coats" in which different types of coat components interact and are co-recruited in the same coated domain (Gopaldass et al, 2024).

The concept of hybrid carriers is compatible with the fact that the spectra of cargos that depend on SNX3 or the SNX-BARs overlap, both in yeast and in mammalian cells (Bean et al, 2017; Steinberg et al, 2013). This might reflect the existence of distinct cargo exit routes that are redundant, perhaps due to similarities in the sorting signals for different SNXs (Strochlic et al, 2007; Ma and Burd, 2019), but it is also consistent with multiple classes of cargos and SNXs jointly using a single endosomal carrier for exiting the endosome. In mammalian cells, evidence for and against a role of common carriers incorporating SNX-BAR and SNX3 cargo has been reported. β2AR, which depends on SNX27 for returning to the plasma membrane, and Wntless, which requires SNX3 for retrograde transport to the Golgi, co-localise on the same endosomes and partition into the same leaving carriers, though with differing enrichment factors (Varandas et al, 2016). On the other hand, knockdown of the Drosophila orthologues of SNX1/SNX2 and SNX5/SNX6, with which SNX27 interacts and cooperates in sorting cargo (Simonetti et al, 2022; Guo et al, 2024; Simonetti et al, 2023; Chandra et al, 2025, 2022), was reported not to influence Wntless function (Harterink et al, 2011). A caveat is that the efficiency of the knockdown was only 80% at the RNA level, and the effects on the protein levels were not tested. The same study found that mammalian Wntless enters only around 20% of SNX-BAR tubules but instead leaves endosomes in smaller SNX3-containing structures.

We set out to probe the potential for generating hybrid coats utilising an in vitro system (Gopaldass et al, 2023) and the sorting nexins Vps5-Vps17 and Snx3, because structures of their respective coats have been solved (Kovtun et al, 2018; Leneva et al, 2021). The in vitro system offers the advantage to monitor coat growth from purified coat components on synthetic supported membrane tubes (SMTubes) in real time in an optically well-defined setting. This allows quantitative assessments of coat radius, speed of growth, and the density of membrane coverage. Here, we exploit this system to show that yeast SNX-BARs and Snx3 readily form adaptable hybrid coats that can incorporate cargos for both classes of sorting nexins and have properties that are distinct from those of homogeneous SNX-BAR or Snx3 coats. We confirm the relevance of these findings in vivo by showing Retromer-mediated interaction between SNX-BARs and Snx3 and by analysing the interdependency of Snx3 and the SNX-BARs for sorting their respective cargos.

## Results

We previously described the formation of SNX-BAR and Retromer-SNX-BAR coats from purified proteins and synthetic supported membrane tubes on the surface of glass slides (SMTubes) (Gopaldass et al, 2023). To extend the system and ultimately enable the analysis of potential hybrid coats, we determined the conditions for Snx3 coat formation on SMTubes using recombinant Snx3 purified from E. coli and Retromer (Vps26-Vps29-Vps35) and SNX-BAR (Vps5-Vps17) complexes purified from yeast (Fig. EV1) (Purushothaman et al, 2017; Gopaldass et al, 2023). As cargo is necessary to stabilise the interaction between Snx3 and Retromer (Leneva et al, 2021; Lucas et al, 2016), we synthesised small peptides that contain the sorting signals and correspond to the cytosolic tails of Ste13 and Ear1, two bona fide Snx3 cargos. The peptides carried an N-terminal HIS₆ tag to recruit them to the membrane via Ni-NTA lipids, and a C-terminal cysteine for maleimide-mediated coupling to fluorescent dyes. The coat proteins were labelled by expressing them as fusion proteins with red or green fluorescing tags, with the tags being fused to Vps17 in the SNX–BAR complex and to Vps29 in Retromer.

### Membrane scaffolding activity and Retromer saturation of Snx3 and SNX-BAR coats

While ^mScarletI^Snx3 bound to the SMTubes in the absence of the Ear1 and Ste13 cargo peptides, it required these peptides to recruit Retromer^mClover and form concentrated domains on the tubules (Fig. 1A). Addition of a SNX–BAR cargo, the Vps10 tail peptide, did not lead to domain formation (Fig. EV2). The intensity of the fluorescent lipid marker Cy5.5.-PE, which had been integrated into the membrane tubules, was reduced underneath the concentrated Snx3 domains. This indicates that the membrane tubes were constricted in these areas relative to the rest of the membrane, where Snx3 and Retromer did not concentrate into a denser arrangement. We hence refer to these constricted regions with a strong concentration of SNX and Retromer as coated domains. We tested whether these Snx3-Retromer domains recruited cargo using

## A

TR-DHPE / SNX-BAR + Retromer-mClover

25nM Retromer
— 21 nm
— 162 nm
— 74 nm

100nM Retromer
— 46 nm
— 57 nm
— 66 nm
— 163 nm

## B

Retromer-mClover fluorescence (a.u.)

○ 25nM Retromer
○ 100nM Retromer

uncoated tube radius (nm)

## C

TR-DHPE / SNX3 + Retromer-mClover

25nM Retromer
— 66 nm
— 80 nm
— 55 nm
— 91 nm

100nM Retromer
— 63 nm
— 78 nm
— 68 nm
— 103 nm

## D

Retromer-mClover fluorescence (a.u.)

○ 25nM Retromer
○ 100nM Retromer

Uncoated tube radius (nm)

## E

add excess
mScarletI-SNX3

GFP-SNX3 + Retromer coats

Excluded from preformed coats

Displacement/substitution into coats

## F

GFP-SNX3 + Retromer  / mScarletI-SNX3

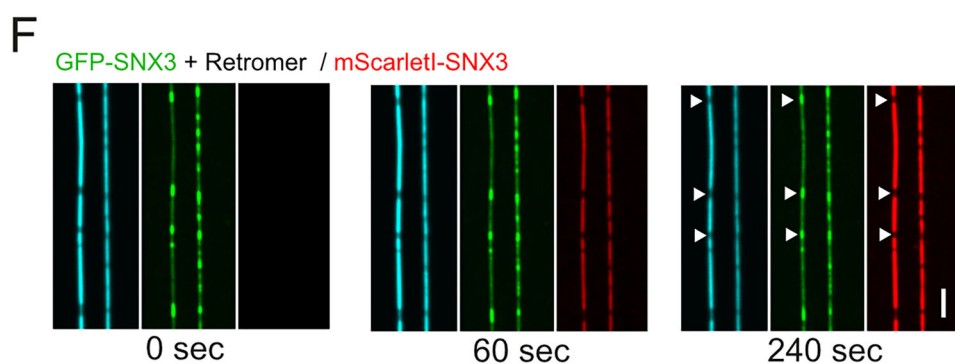

0 sec          60 sec          240 sec

◄

**Figure 2.   Influence of Retromer concentration on SMT constriction and coat formation by Snx3 and SNX-BARs.**

(A) Influence of Retromer concentration on SNX-BAR coat formation. SMTubes labelled with Texas Red DHPE (TR-DHPE) were incubated for 5 min with 25 nM SNX-BAR and 25 nM or 100 nM Retromer[mClover] and imaged by spinning disc fluorescence microscopy. The radius of the non-coated region of each tube is indicated. It was determined as in Fig. 1E. (B) Quantification of Retromer[mClover] fluorescence in coats covering SMTs of various starting radius. The radii of coated and non-coated regions were determined as in Fig. 1. Green: 25 nM SNX-BAR / 25 nM Retromer[mClover] (*n* = 71); yellow: 25 nM SNX-BAR/100 nM Retromer[mClover] (*n* = 97). (C) Influence of Retromer concentration on Snx3 coat formation. SMTubes were incubated for 5 min with 10 μM unlabelled [HIS6]Ear1 cargo peptide, 25 nM Snx3 and 25 nM or 100 nM Retromer[mClover] and imaged by confocal microscopy. The non-coated tube radius is indicated for each tube. (D) Quantification of Retromer[mClover] fluorescence in coats on tubes of various starting radius, determined as in (B). Blue: 25 nM Snx3/25 nM Retromer[mClover] (*n* = 33), Purple: 25 nM Snx3/100 nM Retromer[mClover] (*n* = 50). (E) Scheme of the experiment shown in (F). (F) Lack of subunit exchange in Snx3-Retromer coats. SMTubes were incubated for 2 min with 50 nM [GFP]Snx3/Retromer and 10 μM [HIS6]Ear1 cargo peptide until coats were formed. After a brief wash with buffer to remove the unbound proteins, an excess (100 nM) of [mScarletI]Snx3 was added. Tubes were imaged by spinning disc microscopy for the indicated periods. Scale bars = 2 μm. Source data are available online for this figure.

a fluorescent Alexa-547-labelled version of the Ear1 peptide (Ear1[547]) (Fig. 1B). Coated domains marked by Retromer[mClover] strongly accumulated Ear1[547], confirming that these domains were active for cargo recruitment. All subsequent experiments with Snx3, unless stated otherwise, were performed in the presence of 10 μM of the Ear1 cargo peptide.

SNX-BARs can form dimers and higher-order associations, which at elevated SNX-BAR concentrations can suffice to scaffold a membrane tubule with these proteins alone (Gopaldass et al, 2023; Lopez-Robles et al, 2023; van Weering et al, 2012; Sun et al, 2020; Zhang et al, 2021). By contrast, Snx3 has only a PI3P-binding PX domain and no BAR domain, and it does not show a tendency for spontaneous oligomerisation. Snx3 therefore requires Retromer to form a coat and impose curvature on the membrane (Leneva et al, 2021) (Fig. 1C). Probably for this reason the structure of the Snx3-Retromer coat shows each Snx3 molecule associated with Retromer (Harrison et al, 2014; Leneva et al, 2021), unlike the SNX-BAR-Retromer coat, which contains also SNX-BARs that are not linked to Retromer (Kovtun et al, 2018). We hence compared the amount of Retromer per tubule length in SNX-BAR/Retromer and Snx3/ Retromer coats as a function of the initial membrane tube radius (Fig. 1D,E). While the amount of Retromer[mClover] increased with the starting tube radius for SNX-BAR/Retromer coats (Gopaldass et al, 2023), there was no such tendency for the Snx3/Retromer coats. They entirely failed to constrict wider tubes. The density of Retromer[mClover] in the Snx3/Retromer coats was around twice the density found in the SNX-BAR/Retromer coats (Fig. 1E). We also measured the lipid fluorescence of the tubules under the SNX-BAR and the Snx3 coats (Fig. 1F). This value is directly proportional to the radius of the tubule and can be calibrated by comparing the signals to those under a dynamin oligomer, which compresses the underlying membrane to a known, invariant radius (Roux et al, 2010; Dar et al, 2015). Applying this calibration revealed that Snx3 coats were on average slightly wider than the SNX-BAR coats (Fig. 1G) (20 ± 3.8 nm vs. 19 ± 2.6 nm radius to the outer membrane surface), in line with previous structural studies (Leneva et al, 2021). However, Snx3-Retromer coats were more sensitive to the initial radius of the non-constricted tube than the SNX-BAR-Retromer coats (Fig. 1F). Whereas the radius of SNX-BAR-Retromer coats increased only a little with initial tube radius, the radius of the Snx3-Retromer coat increased more steeply when wider tubes had to be constricted. Thus, the Snx3-Retromer coat may be more adaptable in curvature.

Since the density of Retromer in the Snx3 coat is higher than in the SNX-BAR coat, we tested whether both coats are saturated with Retromer. To this end, we incubated SNX-BAR or Snx3 with either

equimolar concentrations of Retromer[mClover] (25 nM) or with a fourfold excess (100 nM) and measured the amount of Retromer[mClover] incorporated into the coats for tubes of various starting radius (Fig. 2A–D). Similar amounts of Retromer[mClover] were integrated into Snx3 coats, irrespective of whether 25 or 100 nM Retromer[mClover] were offered (Fig. 2C,D). SNX-BAR coats integrated up to twice the amount of Retromer[mClover] when it was present at 100 nM (Fig. 2A,B). Even at this elevated concentration, the density of integrated Retromer remained sensitive to the curvature of the membrane that must be scaffolded. It increased by another factor of two when the coat had to constrict wider membrane tubes. However, the excess of Retromer (100 nM) enabled the SNX-BARs to constrict much wider tubes than the 1:1 mixture (25 nM). This is consistent with Retromer providing driving force for coat formation (Gopaldass et al, 2023), because compressing wider tubes requires more work than compressing thinner tubes. Snx3-based coats behaved very differently. They could oligomerize only on much thinner tubes than the SNX-BAR coats, and even a fourfold excess of Retromer[mClover] did not convey the capacity of constricting significantly wider tubes to them (Fig. 2D).

SNX-BAR-Retromer coats do not incorporate new SNX-BARs or exchange SNX-BARs with the pool in solution (Gopaldass et al, 2023). We also tested this aspect for Snx3-Retromer coats, using a two-stage experiment. In a first stage, [GFP]Snx3-Retromer coats were formed, and non-bound [GFP]Snx3 was washed out. In a second stage, an excess of [mScarletI]Snx3 was flushed into the chamber. This second wave of red fluorescing Snx3 bound readily to the non-coated regions of the tubes but was not incorporated into regions where [GFP]Snx3 had pre-formed coats (Fig. 2E,F). Altogether, our data suggest that both SNX-BAR and Snx3-based Retromer coats are stable and do not readily exchange subunits. Nevertheless, Snx3-Retromer coats can perform less work for membrane scaffolding. They carry a fixed Snx3/Retromer ratio and are saturated with Retromer, unlike the SNX-BAR/Retromer coats, which are more adaptable in terms of their coverage by Retromer.

## SNX-BARs and Snx3 can co-integrate into a hybrid Retromer coat

Since Retromer can interact with both SNX-BARs and Snx3, we tested whether Retromer could form hybrid coats incorporating both classes of sorting nexins. When SMTubes were incubated with equimolar amounts of [mScarletI]Snx3, SNX-BAR[GFP] and Retromer, we observed the formation of hybrid coats harbouring both [mScarletI]Snx3 and SNX-BAR[GFP] (Fig. 3A,B). These hybrid coats could constrict

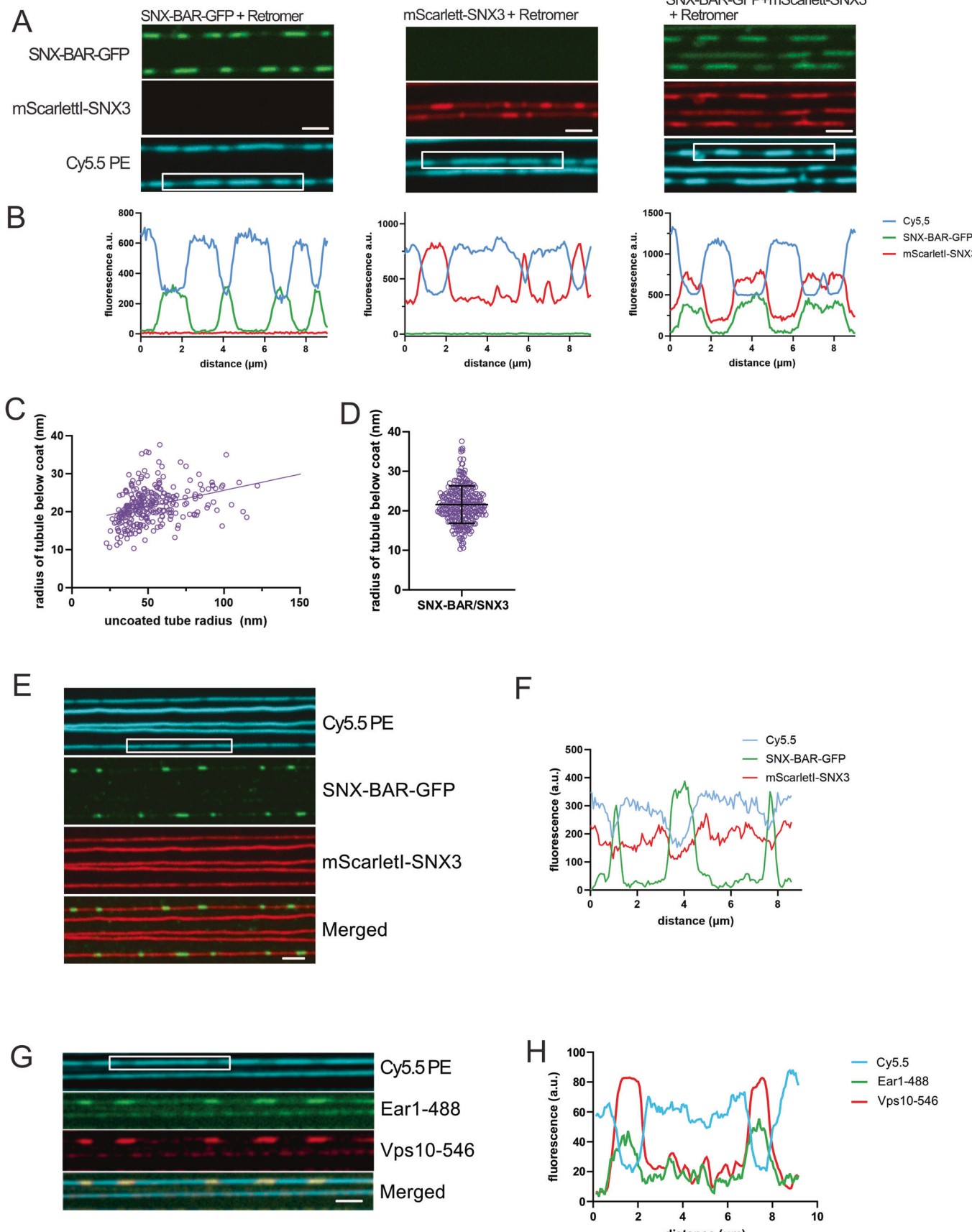

**Figure 3.  Characterisation of hybrid Snx3/SNX-BAR/Retromer coats.**

(A) Co-integration of Snx3 and SNX-BARs. SMTubes were pre-incubated with 10 µM $^{His6}$Ear1 peptide and then incubated with either 25 nM SNX-BAR$^{GFP}$/Retromer, 50 nM $^{mScarletI}$Snx3/Retromer or 25 nM $^{mScarletI}$Snx3/SNX-BAR$^{GFP}$ Retromer (all conditions in the presence of 10 µM $^{HIS6}$Ear1) for 3–5 min and imaged by spinning disc microscopy. (B) Intensity scans along the tubes boxed in (A). (C) Quantification of the radius under hybrid coats as a function of the starting tube radius ($n = 275$). (D) Cumulative plot of the data in (C), disregarding the uncoated tube radius. $n = 158$. Bars represent the means and the standard deviation. (E) Lack of hybrid coat formation in the absence of Retromer. SMTubes were incubated with 100 nM of SNX-BAR$^{GFP}$, 100 nM of $^{mScarletI}$Snx3 and 10 µM $^{His6}$Ear1 peptide as in A, but without Retromer. (F) Intensity scan along the tube sections boxed in (E). (G) Co-integration of Snx3 and SNX-BAR cargo. SMTubes were pre-incubated with 5 µM $^{His6}$Ear1-488 and 5 µM $^{His6}$Vps10-546. After 10 min of incubation with the cargo peptides, 25 nM each of Snx3, SNX-BAR and Retromer were added. After 3–5 min of incubation, unbound proteins were washed away, and tubes were imaged by confocal microscopy. (H) Intensity scan along the boxed tube section shown in (C). All scale bars: 2 µm. Source data are available online for this figure.

tubules of up to 125 nm initial radius (Fig. 3C), resembling in this respect the SNX-BAR coats (Fig. 1F). However, the hybrid coats were larger, with a radius of 22 ± 4.7 nm (Fig. 3D), compared to 19 ± 2.6 nm for homogeneous SNX-BAR-Retromer and 20 ± 3.8 nm for homogeneous Snx3-Retromer coats (Fig. 1G). Hybrid coat formation depended on Retromer. Co-incubating SMTubes with SNX-BAR$^{GFP}$ and $^{mScarletI}$Snx3 without Retromer led to the formation of coats containing only SNX-BARs but not enriching Snx3 in this zone (Fig. 3E,F). This suggests that Snx3 is not recruited into hybrid coats by a direct interaction with SNX-BARs but requires Retromer for this. Using fluorescently labelled Ear1 and Vps10 peptides revealed that the Retromer hybrid coats concentrate both peptides (Fig. 3G,H) and are hence competent for cargo integration.

The structures of Snx3 or SNX-BAR-coated membrane tubules show limited regularity of these coats and some voids, which might provide space to incorporate other proteins (Leneva et al, 2021; Kovtun et al, 2018). We hence performed staged experiments to test whether Snx3 could integrate into pre-formed SNX-BAR-Retromer coats or vice versa (Fig. 4A–D). SMTs were incubated with 25 nM SNX-BAR$^{GFP}$ and Retromer to allow the formation of coats that were 1–2 µm long. Unbound proteins were washed away, and an excess (100 nM) of $^{mScarletI}$Snx3 was added (Fig. 4A,B). After 5 min of incubation, $^{mScarletI}$Snx3 was mainly localised to the uncoated regions, but it did not integrate into the pre-formed SNX-BAR-Retromer coats. Similarly, pre-formed $^{mScarletI}$Snx3-Retromer coats could not integrate an excess of SNX-BAR$^{GFP}$ (Fig. 4C,D). These observations suggest that the pre-formed homogeneous coats do not contain suitable voids and binding sites and/or are not flexible enough to integrate the other sorting nexin class.

To directly compare SNX densities in the hybrid and homogeneous coats, we created a $^{GFP}$Snx3 construct using the same GFP tag as in our SNX-BAR$^{GFP}$. Retromer coats were generated with either $^{GFP}$Snx3 or SNX-BAR$^{GFP}$ in the presence or absence of non-tagged versions of the respective other sorting nexin (Fig. 5A,B). The overall sorting nexin concentration was kept constant (50 nM for pure coats and 25 nM of each sorting nexin class for hybrid coats, both in the presence of 50 nM Retromer) (Fig. 5A–C). The mean GFP fluorescence per tube length in the $^{GFP}$Snx3/Retromer coat (404 ± 80 a.u., $n = 40$) was about twice the mean GFP fluorescence in the SNX-BAR$^{GFP}$ coat (230 ± 40 a.u., $n = 40$) (Fig. 5C). Considering that the SNX-BAR complex is a heterodimer of a labelled Vps17$^{GFP}$ and an unlabelled Vps5 it follows that the SNX densities in Snx3-based and SNX-BAR-based Retromer coats are similar.

The different degrees of labelling of $^{GFP}$Snx3 (every subunit is labelled) and SNX-BAR$^{GFP}$ (a heterodimer, in which only Vps17$^{GFP}$

is labelled, but not Vps5) could also be exploited in replacement experiments. To this end, Retromer coats were generated at different ratios of $^{GFP}$Snx3 over SNX-BAR$^{GFP}$ (Fig. 5D,E). The total GFP signal in the coats should remain constant if one $^{GFP}$Snx3 molecule replaced a SNX-BAR dimer, but it should increase if two Snx3 molecules replaced a SNX-BAR dimer. Regarding a coat region of 6 sorting nexins as an example, this unit would contain 3 GFP when made only of SNX-BAR$^{GFP}$ heterodimers. In a coat assembled at a SNX-BAR$^{GFP}$ : $^{GFP}$Snx3 ratio of 2:1, this unit would contain 4 GFPs (2 from Vps17$^{GFP}$, 2 from $^{GFP}$Snx3); at a 1:2 ratio it would contain 5; and in a pure $^{GFP}$Snx3 coat it would contain 6. The fluorescence signals of hybrid coats that assembled from $^{GFP}$Snx3 and SNX-BAR$^{GFP}$ at these ratios followed exactly this expected pattern (Fig. 5D,E). This result suggests that one Snx3 substitutes one SNX-BAR subunit and that hybrid coats can assemble at variable SNX-BAR/Snx3 stoichiometries. This latter point was further supported when we compared coats formed from mixtures containing $^{GFP}$Snx3 and unlabelled SNX-BARs at ratios of either 2:0, 2:1 or 1:2. Retromer was kept at the same concentration in all cases (Fig. 5F–H). The Snx3$^{GFP}$ signal in the coat decreased in proportion to the SNX-BAR:Snx3$^{GFP}$ ratio in solution. Hybrid coats thus do not have a fixed composition, suggesting that they are adaptable as a function of available Snx3 and SNX-BARs and/or their cargos.

## Hybrid coat formation enhances the capacity for membrane scaffolding

We noticed that hybrid coats could form on and constrict tubes of higher starting radius than homogeneous Snx3 coats (Fig. 5B,D,G), suggesting that mixed coats are more potent in deforming the membrane and that SNX-BARs might help drive Snx3/Retromer coat formation. This points to a potential synergy between Snx3 and the SNX-BARs in scaffolding the tubes. We used giant unilamellar vesicles (GUVs) to test whether this is the case. On GUVs, coat formation requires more work for membrane deformation because a tubular membrane conformation is not pre-defined as in the case of SMTubes. Tubulation also reduces the volume of the GUV and necessitates extrusion of luminal liquid, requiring additional work. When GUVs were incubated with 100 nM $^{mScarletI}$Snx3 and Retromer, tubules did not form (Fig. 6A,B). Substituting a quarter of the Snx3 by SNX-BARs (25 nM SNX-BAR$^{GFP}$:75 nM Snx3) resulted in vigorous tubule formation on about 60% of the GUVs. This tubulation was stronger than the tubulation observed with a 25 nM SNX-BAR-GFP or with 100 nM $^{mScarletI}$Snx3 alone. We observed a similar effect when cargos for both Snx3 and SNX-BAR were present simultaneously (Fig. EV3A,B).

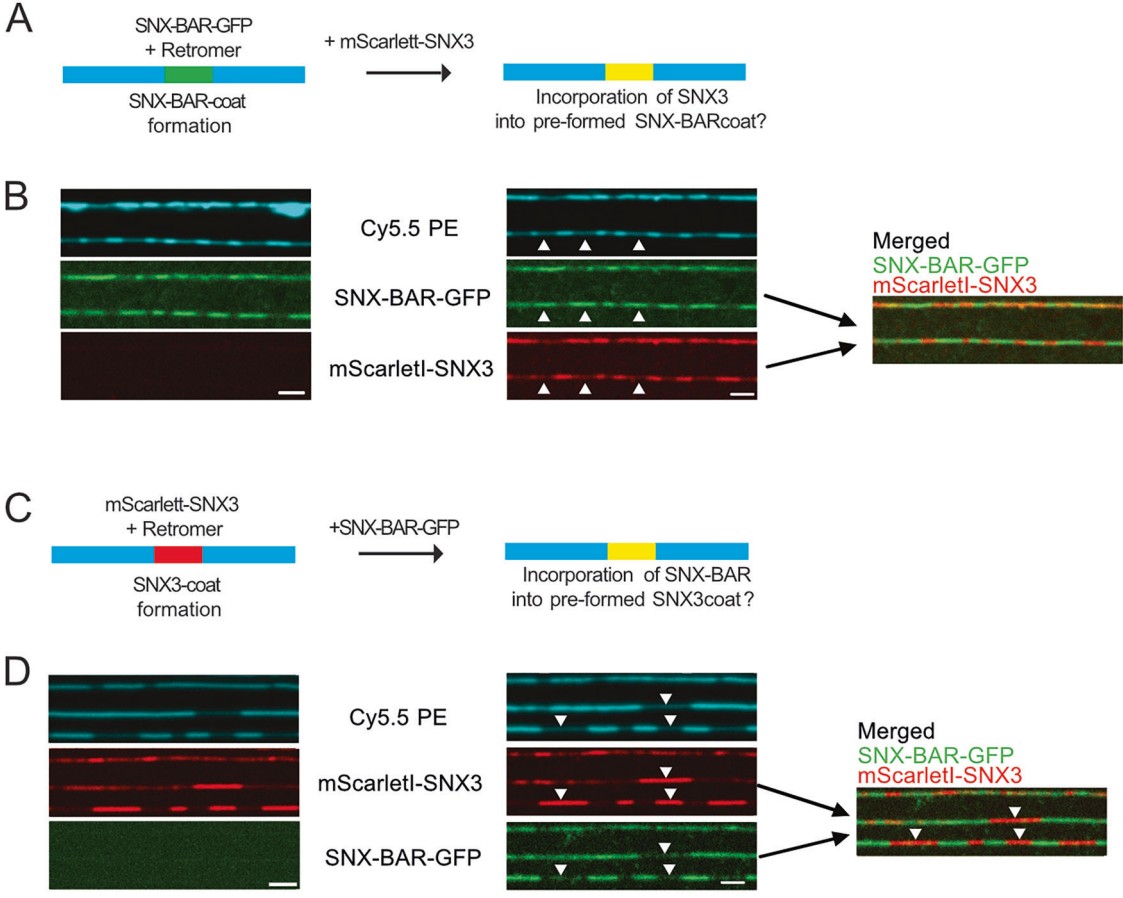

**Figure 4. Homogeneous Snx3 or SNX-BAR coats cannot integrate the other sorting nexin class.**

(A) Scheme of the experiment shown in (B). (B) Non-integration of Snx3 into pre-formed SNX-BAR Retromer coats. SMTubes were incubated with 25 nM of SNX-BAR[GFP]/Retromer for 2–3 min until coats were formed. After a brief wash to remove unbound proteins, 100 nM [mScarletl]Snx3 was added and incubated for 5 min. Tubes and coats were imaged by confocal microscopy. Arrowheads point to SNX-BAR[GFP]/Retromer coats, which are not accessible to [mScarletl]Snx3. (C) Scheme of the experiment shown in (D). (D) Non-integration of SNX-BARs into pre-formed Snx3 Retromer coats SMTubes were incubated with 50 nM of [mScarletl]Snx3/Retromer for 2–3 min until coats were formed. After a brief wash with buffer, 50 nM SNX-BAR[GFP] was added and incubated for 5 min. Tubes and coats were imaged by confocal microscopy. Arrowheads point to [mScarletl]Snx3/Retromer coats, which are not accessible to the excess of added SNX-BAR[GFP]. Scale bars = 2 μm. Source data are available online for this figure.

Thus, hybrid coat formation allows Snx3 and SNX-BARs to provide higher membrane scaffolding activity than in respective homogeneous coats.

## Correlates of hybrid coat formation in vivo

To search for in vivo correlates of hybrid coat formation, we first compared the localisation of Snx3 and SNX-BARs by tagging Snx3, Vps5 or Vps17 with either mNeonGreen (mNG) or yomCherry (Fig. 7A,B). The major pool of Snx3[mNG] colocalized with Vps5[yomCherry] (Fig. 7A). To have a reference point, we performed the same experiment with Vps5[yomCherry] and Vps17[mNG] as these proteins form a stable heterodimer (Seaman et al, 1998; Seaman and Williams, 2002) and can therefore provide a benchmark for maximal colocalization (Fig. 7A). The Pearson correlation coefficients showed similar colocalization between Snx3[mNG] and Vps5[yomCherry] as between Vps5[yomCherry] and Vps17[mNG] (Fig. 7B). Ablation of Vps35, which is necessary for forming hybrid and homogeneous Snx3 coats in vitro (see above), reduced

colocalization of Snx3[mNG] and Vps5[yomCherry] and displaced Snx3[mNG] mainly to the vacuolar membrane and the cytosol. Although the resolution of these experiments does not permit to resolve individual tubules and visualise hybrid coat formation in situ directly, the result is at least consistent with this concept because it relates the co-existence of SNX-BAR and Snx3 in the same compartment to coat formation.

To address this point more directly, we assayed the interaction of Snx3 and SNX-BARs. The binding sites of Snx3 and SNX-BARs on Retromer overlap (Leneva et al, 2021; Kovtun et al, 2018), rendering the simultaneous binding of Snx3 and SNX-BARs to a single Retromer complex unlikely. However, Retromer forms arch-like dimers, which might permit binding of SNX-BARs at one end and of Snx3 at the other. To test this, we generated a strain carrying tagged SNX-BAR (Vps5[HA]), Snx3 (Snx3[V5]) and Retromer (Vps35[yomCherry]). We also generated a *vps35* mutant impaired in Snx3 binding (*vps35[QGRE]*, with the substitutions Q181A, G182A, R185A, and E186A). These residues were determined by combining information from previous chemical cross-linking studies

A

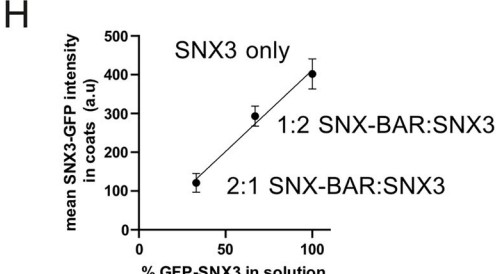

B

C

SNX density on coats

D

E

F

G

H

**Figure 5.  Sorting nexin stoichiometry in hybrid coats.**

(A) Comparison of Snx3 and SNX-BAR densities in hybrid and the respective homogeneous coats. SMTubes were incubated with Retromer and the following combinations of sorting nexins carrying a GFP-tag either on Snx3 or the SNX-BARs: 50 nM [GFP]Snx3 and 50 nM Retromer 25 nM [GFP]Snx3 with 25 nM SNX-BARs and 50 nM Retromer, 25 nM of the SNX-BAR[GFP] heterodimer (25 nM Vps5 and 25 nM Vps17[GFP]) and 25 nM Retromer and 25 nM Snx3 with 25 nM SNX-BAR[GFP] and 50 nM Retromer. After 2–5 min of incubation, the SMTubes were imaged by spinning disc microscopy. (B) Quantification (performed as in Fig. 1) of the GFP intensity for coats on tubes of various starting radius, comparing homogeneous SNX coats with coats formed from mixtures of the sorting nexins. [GFP]Snx3/Retromer (n = 22), SNX-BAR[GFP]/Retromer (n = 40), [GFP]Snx3/SNX-BAR/Retromer (n = 25), Snx3/SNX-BAR[GFP]/Retromer (n = 84) (C). The mean and standard deviation of the [GFP]SNX intensity were calculated for each type of coat from the data in (A, B). (D) Integration of sorting nexins at variable stoichiometries. SMTubes were incubated with 50 nM Retromer and sorting nexins at four different ratios of [GFP]Snx3 to SNX-BAR[GFP]: 1:0 = 50 nM [GFP]Snx3 only; 2:1 = 25 nM [GFP]SNX3 and 12.5 nM SNX-BAR[GFP]; 1:2 = 12.5 nM [GFP]SNX3 and 25 nM SNX-BAR[GFP], and 0:1 = 25 nM SNX-BAR[GFP]. After coat formation, unbound protein was washed away with PBS buffer, and tubes were imaged and quantified as in (B). (E) The mean and standard deviation of the total [GFP]SNX intensity were calculated for each type of coat from the data in (D). Number (n) of scored items: [GFP]SNX3 (n = 41); 1:2 SNX-BAR[GFP]:[GFP]SNX3 (n = 60); 2:1 SNX-BAR[GFP]:[GFP]SNX3 (n = 57); SNX-BAR[GFP] (n = 63). (F) Comparison of the [GFP]Snx3 intensity in coats formed in the presence of various ratios of competing SNX-BARs. 1:0 = 25 nM [GFP]Snx3 only; 1:2 = 12.5 nM SNX-BAR[GFP] and 25 nM [GFP]Snx3 and 2:1 = 25 nM SNX-BAR[GFP] and 12.5 nM [GFP]Snx3 (G) Quantification (as in Fig. 1) of the data shown in (F). Snx3 (n = 24), 1:2 SNX-BAR:Snx3 (n = 65), 2:1 SNX-BAR:Snx3 (n = 83). (H) The mean [GFP]Snx3 intensity and the standard deviation of the data in (E, F), shown as a function of % of total SNXs in solution. Scale bars: 2 μm. Number (n) of scored items: [GFP]SNX3 (n = 24); 2:1 [GFP]SNX3:SNX-BAR (n = 65); 1:2 [GFP]SNX3:SNX-BAR (n = 83). Source data are available online for this figure.

(Harrison et al, 2014) with Alphafold 3 modelling and an assessment of their contribution to the interaction interface using the PDB-PISA software (Fig. EV4). We used immunoprecipitation assays to probe whether Snx3 and SNX-BARs interact through Retromer. In the immunoprecipitation experiments, wild-type Vps35[yomCherry] pulled down both Snx3[V5] and Vps5[HA]. By contrast, Vps35[QGRE-yomCherry] lost the interaction with Snx3[V5] but maintained the interaction with Vps5[HA] (Fig. 7C,D). When we pulled down Snx3[V5] from VPS35[yomCherry] wild-type cells, both Vps35[yomCherry] and Vps5[HA] co-adsorbed to the beads (Fig. 7E,F). The Snx3[V5]-Vps5[HA] interaction was reduced in the *vps35[QGRE]* background. This suggests that Retromer can bridge Snx3 and SNX-BARs.

To examine the in vivo relevance of this association, we tagged yeast Retromer cargo proteins and followed their fate in vivo (Fig. 8). We used Vps10 as a bona fide SNX-BAR cargo (Bean et al, 2017; Purushothaman and Ungermann, 2018; Suzuki et al, 2019), and Ear1 and Ste13 as Snx3 cargos. In wild-type cells, a Vps10[mNG] fusion protein localised to cytosolic puncta which, based on previous work (Suzuki et al, 2019; Day et al, 2018), should represent yeast endosome/late Golgi (Fig. 8A). In SNX-BAR knockout cells (*vps5Δ* or *vps17Δ*), Vps10[mNG] colocalised with the vacuolar marker FM4-64. This vacuolar localisation results from a failure to recycle Vps10 to the Golgi (Nothwehr et al, 1999; Cereghino et al, 1995). In *snx3Δ* knockout cells, Vps10[mNG] localisation was moderately affected, as evident from cells showing weak staining of the vacuolar membrane with Vps10[mNG], in line with previous results (Bean et al, 2017). The Increased colocalization between Vps10[mNG] and the vacuolar lipid marker FM4-64 confirmed this limited mislocalization to the vacuolar membrane (Fig. 8B). Furthermore, the *vps35[QGRE]* mutant, which targets the Retromer-Snx3 interaction, showed a partial accumulation of Vps10[mNG] on the vacuolar membrane, suggesting that the recruitment of Snx3 to Retromer is relevant for Vps10 sorting (Fig. EV5). Vice versa, the Snx3 cargos Ear1[mNG] (Fig. 8C,D) and Ste13[mNG] (Fig. 8E,F) showed significant mislocalization to vacuoles not only in *snx3Δ*, but also in *vps5Δ* or *vps17Δ* cells, suggesting that the Snx3 recycling pathway requires SNX-BARs for its function. We extended these observations to two other Snx3 cargos, Kex2 and Pep12 (Voos and Stevens, 1998; Hettema et al, 2003), with similar results (Fig. EV6). These in vivo observations are in line with our in vitro findings and support the notion that Retromer can link SNX-BARs and Snx3 into hybrid

coats that promote recovery of cargos for both classes of sorting nexins.

## Discussion

Structural studies revealed many features of Retromer associated with either the SNX-BAR Vps5 or Snx3 (Leneva et al, 2021; Kovtun et al, 2018; Lucas et al, 2016; Kendall et al, 2020, 2022). These coats share the organisation of Retromer dimers into arches that cross-link a SNX layer on a membrane tubule. The available structures support the current view that each SNX that associates with Retromer defines a separate sorting route. However, larger scale analyses of the impact of various endosomal coat components on the steady state distribution of a broad spectrum of cargo have revealed a significant overlap in the cargos affected by different classes of sorting nexins, including Snx3 and SNX-BARs, both in yeast and in mammalian cells (Bean et al, 2017; McNally et al, 2017; Steinberg et al, 2013). Further observations exist that are consistent with the formation of hybrid coats between Snx3 and SNX-BARs, but were not explicitly interpreted from this perspective. In mammalian cells, Retromer cargo for SNX3 (Wntless) and the SNX-BAR-associated protein SNX27 (β2-adrenergic receptor) (Harterink et al, 2011; Simonetti et al, 2022) were partially found in and left from the same endosomal sorting domains, although they use different sorting nexins and have different destinations (Varandas et al, 2016). The sorting signals of Wntless and β2-adrenergic receptor could be exchanged without impairing their entry into endosomal sorting domains (Varandas et al, 2016). Also in yeast, Snx3 and Vps17 were reported to partially co-localise, and both were necessary for efficient sorting of the iron transporter Ftr1 (Strochlic et al, 2007). Another link is provided by overexpression of the Snx3 cargo Ear1, which led to missorting of the SNX-BAR cargo Vps10 to the vacuole, and by the observation that Ear1 retrieval was compromised in a *vps5Δ* mutant (Suzuki et al, 2019). These studies were not designed to exclude indirect effects and were hence not interpreted in the context of hybrid coat formation. Therefore, models linking one type of homogenous sorting nexin coat with a defined population of cargo and a transport direction have prevailed in the general discussion of endosomal protein sorting. Our direct observation of the hybrid coat formation from

## A

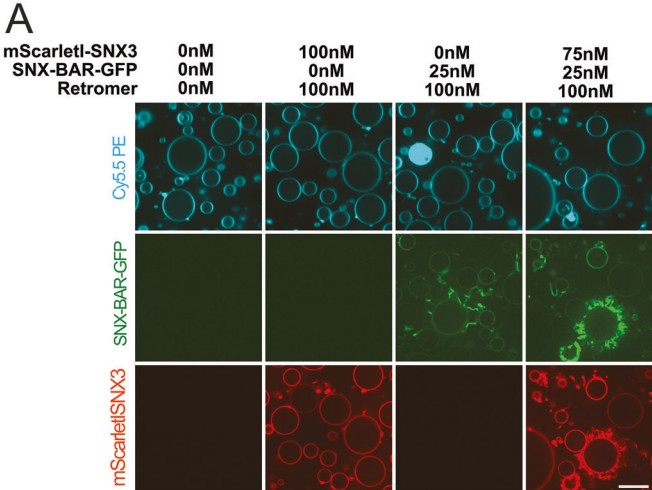

## B

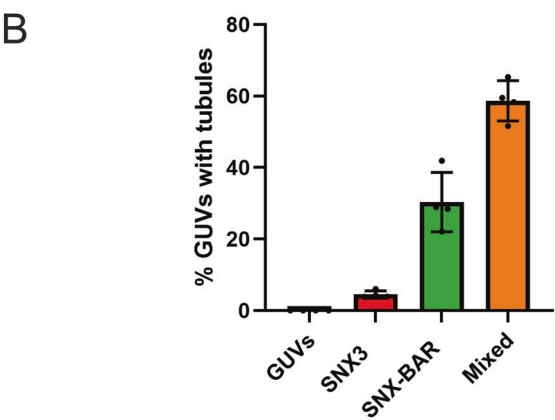

**Figure 6. Tubulation of GUVs by hybrid coats.**

(A) GUVs labelled with Cy5.5-PE were incubated with the indicated concentrations of Retromer, ^mScarletI^Snx3, and SNX-BAR^GFP^, and 10 μM Ear1 peptide. After 1 h of incubation, the GUVs were imaged by confocal microscopy. (B) Quantification of the percentage of GUVs with tubules for each condition shown in (A). Four independent experiments were quantified. Total number of scored items: GUVs only ($n = 186$); Snx3/Retromer ($n = 177$); SNX-BAR/Retromer ($n = 193$); Snx3/SNX-BAR/Retromer ($n = 358$). Scale bar: 5 μm. Source data are available online for this figure.

purified proteins now provides a physical correlate and a potential explanation for the overlaps in the coat dependencies of cargo transport.

The hybrid coats assembled from Snx3 and SNX-BARs that we have analysed can integrate cargo peptides for both sorting nexin classes, suggesting that they are functional for cargo collection. However, they have properties that are distinct from those of the respective homogeneous coats. While both Snx3 and SNX-BARs oligomerize with Retromer into stable coats on SMTubes, Snx3/Retromer cannot constrict wider tubules, which SNX-BAR-Retromer readily scaffolds. Such differences could of course be due to factors such as the lipid composition of endosomal membranes, membrane tension, as well as the dynamics of coat formation, which may not be the same for the two coats. A more

parsimonious, and in our view more likely explanation, is that the scaffolding potential of the two coats reflects their different capacity to self-interact. SNX-BARs self-interact and form a lattice on their own that can deform the membrane, and Retromer adds additional bonds and driving force to this process (Gopaldass et al, 2023; Simunovic et al, 2015; Lopez-Robles et al, 2023). In a Snx3 coat, however, the sorting nexins do not self-interact, and Retromer is the only element that crosslinks them and imposes curvature on the membrane (Leneva et al, 2021; Kovtun et al, 2018). Integration into hybrid coats apparently compensates for this lack of scaffolding activity of Snx3, allowing it to populate and constrict tubes of similar width as a homogeneous SNX-BAR coat, and to tubulate GUVs that it would otherwise not be able to deform.

Available structures of Snx3- and SNX-BAR Retromer coats suggest that simultaneous binding of the two sorting nexin classes to Retromer might be impossible due to steric clashes (Leneva et al, 2021; Kovtun et al, 2018; Lucas et al, 2016). Hybrid coats could thus form in two ways, which are not mutually exclusive. They might contain interspersed homogeneous zones that are covered only by Snx3-Retromer or SNX-BAR-Retromer and are too small to be resolved by our light-microscopic analysis. Alternatively, Retromer oligomers, which can form through dimerisation of Vps35 or Vps26 (Lucas et al, 2016; Leneva et al, 2021; Kovtun et al, 2018; Kendall et al, 2022, 2020), could cross-link Snx3 and SNX-BARs into a joint, oligomeric structure. This latter possibility is favoured by the Retromer-dependent interaction of Snx3 with the SNX-BARs that we have detected.

The sorting nexin densities of homogeneous Snx3- and SNX-BAR-based coats on SMTs are comparable. After their formation, these homogeneous coats do not readily exchange subunits. They cannot integrate sorting nexins from the respective other class, suggesting that they are stable structures and that they do not contain suitable voids to convert into hybrid coats. The composition of the coats is apparently defined when they are being formed. During formation, however, there is considerable adaptability. Hybrid coats can form at variable Snx3/SNX-BAR ratios. Our replacement experiments suggest that this leaves the overall sorting nexin density unchanged, implying that SNX-BAR and Snx3 subunits occupy similar average membrane areas in the coats. The adaptability of hybrid coat composition is an interesting feature, particularly in combination with a role for cargo in coat formation. Snx3, for example, can bind to membranes by itself, but its integration into coats requires cargo (Fig. 1)(Lucas et al, 2016; Leneva et al, 2021). An adaptable coat stoichiometry combined with cargo dependence could offer a simple mechanism for adjusting coat composition to the amounts of cargos for Snx3 and SNX-BAR that must be transported.

Hybrid coats can have advantages. For example, they enhance the scaffolding activity of Snx3, enabling coat formation on less curved membranes that do not support a homogeneous Snx3–Retromer coat, presumably because such a coat cannot provide the required driving force. However, the formation of hybrid coats also has important conceptual implications. We generally assume that cargo recruitment into a homogeneous carrier—mediated by interactions between targeting motifs and coat components—is the major factor determining the cargo's final destination. When cargos with different destinations enter the same

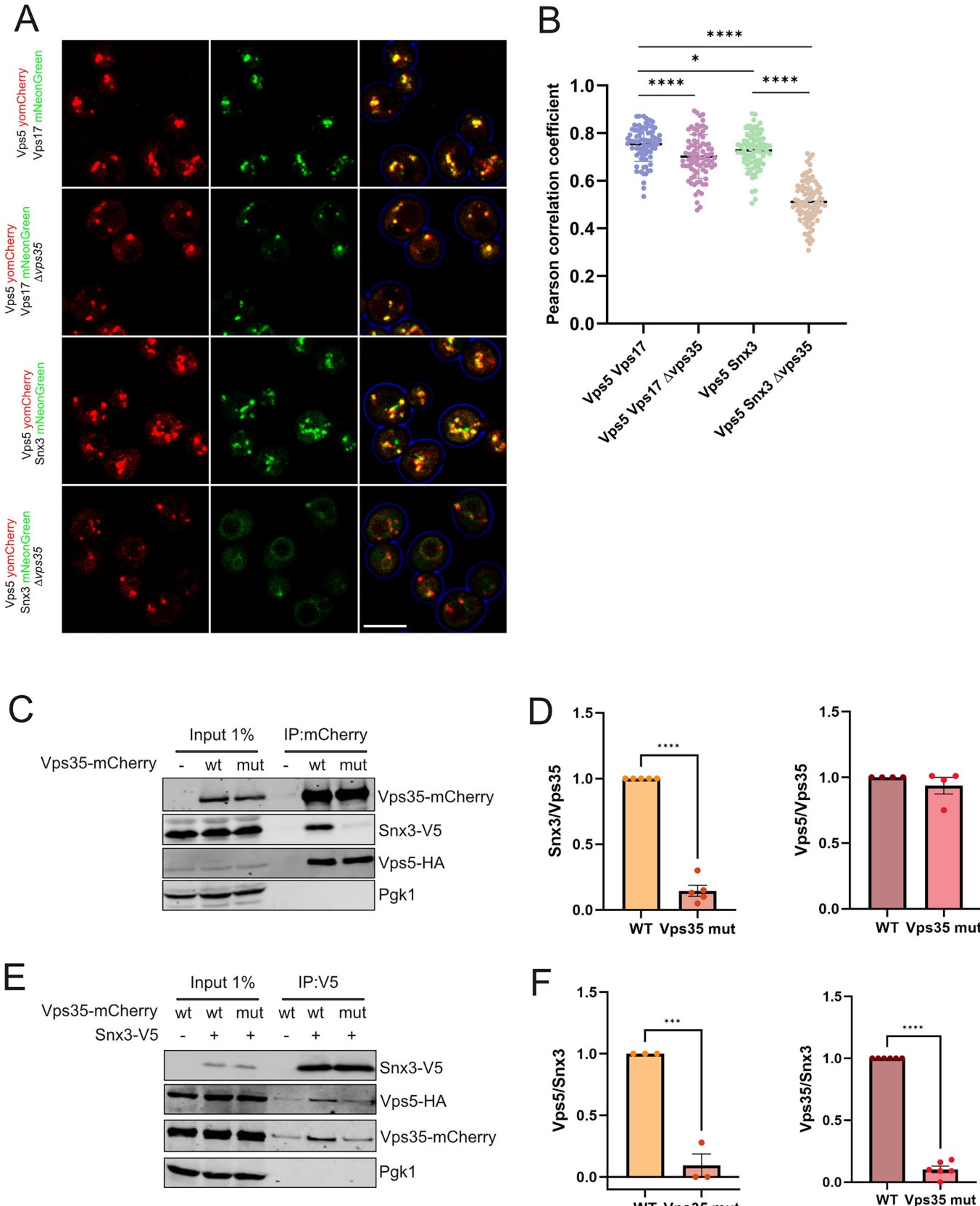

**Figure 7. Snx3 and SNX-BARs interact in vivo depending on Retromer.**

(A) Colocalization of Snx3 and SNX-BARs. Wild-type or *vps35Δ* cells co-expressing Vps5$^{yomCherry}$ and Vps17$^{mNG}$, or Vps5$^{yomCherry}$ and $^{mNG}$Snx3 were analysed by a z-series of confocal images taken at a z-distance of 0.3 μm. Maximum projections were generated in ImageJ. Scale bar: 5 μm. (B) Pearson correlation coefficient (PCC) values were calculated using the JACoP plugin in the ImageJ software for a total of >80 cells from three independent experiments. See Fig. EV7A for biological replicate variability. Bars represent the mean values and standard deviation. A two-tailed unpaired *t* test was used to evaluate significance. ****$P < 0.0001$; ***$P < 0.001$; **$P < 0.01$ and *$P < 0.1$. (C) Immunoprecipitation of mCherry-tagged Vps35$^{wt}$ or Vps35$^{QGRE}$. Logarithmically growing *vps35Δ* cells expressing Vps35$^{mCherry}$ (WT) or Vps35$^{QGRE-mCherry}$ (Mut) from an integrative plasmid, and genomically tagged Snx3$^{V5}$ and Vps5$^{HA}$, were lysed in detergent. Proteins were pulled down with an affinity matrix to mCherry and analysed by SDS-PAGE and Western blotting against the indicated tags. (D) Quantification of the amount of Vps5 and Snx3 pulled down by mCherry-labelled Vps35$^{wt}$ or Vps35$^{QGRE}$. The ratios between the signals for Vps35 and Snx3 or Vps5 were calculated. For Vps35$^{WT}$ the ratio was set to 1 as a reference. Mean and standard deviation from five independent experiments are shown. A two-tailed unpaired *t* test was used to evaluate significance. ****$P < 0.0001$; ***$P < 0.001$; **$P < 0.01$ and *$P < 0.1$. (E) Snx3$^{V5}$ was pulled down in an experiment as in (C), using *vps35Δ* cells expressing Vps35$^{mCherry}$ (WT) or Vps35$^{QGRE-mCherry}$ (Mut) from a plasmid and genomically tagged Snx3$^{V5}$ and Vps5$^{HA}$ in the indicated combinations. (F) Quantification of Vps35$^{mCherry}$ and Vps5$^{HA}$ pulled down by Snx3$^{V5}$. The ratios between the signals for Snx3$^{V5}$ and Vps35$^{mCherry}$ or Vps5$^{HA}$ were calculated. For Vps35$^{wt}$, the ratio was set to 1 as a reference. Mean and standard deviation from three to five independent experiments are shown. A two-tailed unpaired *t* test was used to evaluate significance. ****$P < 0.0001$; ***$P < 0.001$; **$P < 0.01$ and *$P < 0.1$. Source data are available online for this figure.

hybrid carrier, targeting must instead occur after carrier formation through additional mechanisms.

Several possibilities have been proposed (Varandas et al, 2016). These may include correction through subsequent trafficking steps that remove misdelivered cargo or selective retention at the correct location, for example, through interactions with resident proteins. Biophysical properties of the cargo may also contribute to this retention, such as the length of its transmembrane domain. Organelles and the plasma membrane exhibit distinct, characteristic membrane thicknesses; therefore, the transmembrane domain length must match the compartment in which the cargo resides (Sharpe et al, 2010; Watson and Pessin, 2001; Cosson et al, 2013).

Taken together, these considerations lead to a concept in which a cargo protein's sorting signal and its targeting information become partially decoupled. The sorting signal may function primarily as an entry ticket into a carrier without necessarily specifying the cargo's ultimate destination. It may increase the efficiency of recruitment into a coat—serving as an accelerator of transport along a trafficking route—whereas the final location would be determined by a more complex set of interactions and compatibility cues. This final destination might be reached only after several rounds of trafficking and "error correction," which ultimately leads the cargo to the correct compartment, where it encounters the most favourable set of matching interactions for its retention.

We consider it likely that additional combinations of hybrid coats exist. This may apply to Retromer, because in yeast Snx3 and Snx4 substantially co-localise with the SNX-BAR protein Vps17 (Strochlic et al, 2007). More broadly, it is conceivable that other endosomal coat components may also co-enrich on tubules generated at the surface of endosomes. Many such components are sorting nexins with affinities for the highly curved environment of tubular membranes (Teasdale and Collins, 2012; van Weering et al, 2010; Solinger and Spang, 2022; Shortill et al, 2022). Thus, once tubules are generated—e.g. through the activity of motors or actin polymerisation—they may attract a mixture of coat components.

Testing this possibility will require careful live-imaging approaches both in vivo and in vitro, as well as continued structural analyses of coat complexes. This will be particularly interesting in metazoan cells, which use a wider and more complex array of coat systems than yeast, that can exhibit different or additional properties.

## Methods

### Reagents and tools table

| Reagent/resource | Reference or source | Identifier or catalogue number |
|---|---|---|
| **Experimental models** | | |
| Saccharomyces cerevisiae SEY6210 | ATCC | 96099 |
| **Antibodies** | | |
| RFP-Trap Magnetic Particles | Chromotek | rtd |
| V5-Trap Magnetic Agarose | Chromotek | v5ta |
| V5-tag Polyclonal antibody | Chromotek | 14440-1-AP |
| mCherry Polyclonal antibody | Chromotek | 26765-1-AP |
| Anti-HA tag antibody | ABCAM | Ab137838 |
| **Oligonucleotides and other sequence-based reagents** | | |
| All primers | Microsynth, Switzerland | n/a |
| **Chemicals, enzymes and other reagents** | | |
| Egg L-alpha-phosphatidylcholine (EPC) | Avanti lipids | 840051 |
| 1,2-dioleoyl-sn-glycero-3phospho-L-serine sodium salt (DOPS) | Avanti lipids | 840035 |
| 1,2-dioleoyl-sn-glycero-3phospho-(1'-myo-inositol-3'-phosphate) (PI3P) | Avanti lipids | 850150 |
| Red DHPE | Thermofisher | T1395MP |
| FM4-64 | Sigma-Aldrich | SCT127 |
| 3-Glycidyloxypropyltrimethoxysilane | Sigma-Aldrich | 440167 |
| **Software** | | |
| PRISM | GraphPad | |
| **Other** | | |
| μ-Slide VI 0.4 | IBIDI | 80606 |
| Aladdin Single-Syringe Pump | World Precision Instruments | AL-1000 |

## Materials

Lipids were purchased from Avanti Polar Lipids (USA): Egg L-alpha-phosphatidylcholine (EPC); 1,2-dioleoyl-sn-glycero-3-phospho-L-serine sodium salt (DOPS); 1,2-dioleoyl-sn-glycero-3-

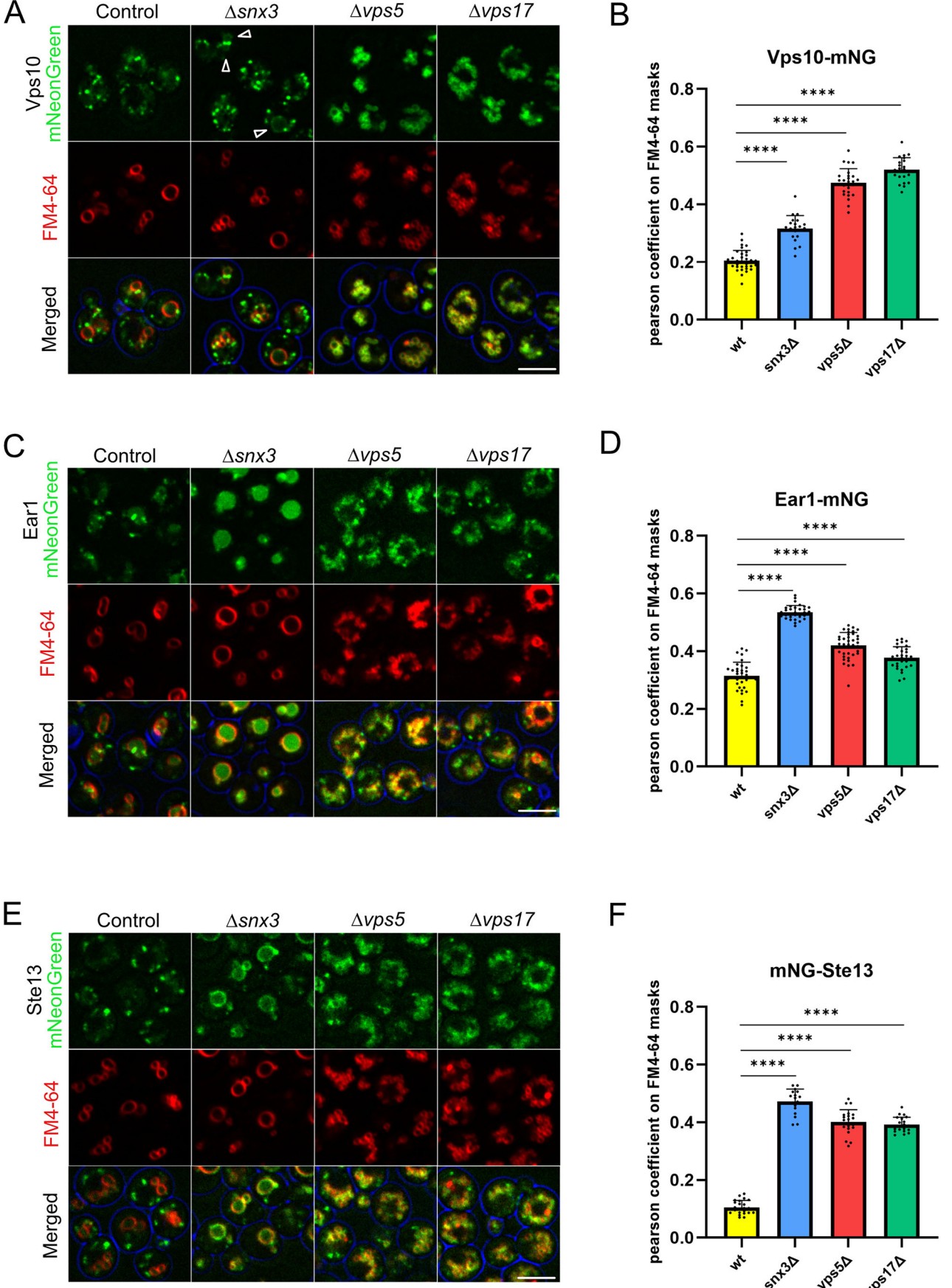

◄  **Figure 8.   Mutual dependence of Snx3 and SNX-BARs for cargo recycling.**

(A) Vps10 distribution. Logarithmically growing cells expressing Vps10^mNG in wild-type, *Snx3Δ*, *vps5Δ*, and *vps17Δ* background were stained with FM4-64 and analysed by confocal microscopy. Average intensity projections of confocal z-stacks taken at a z-interval of 0.3 μm are shown. Arrowheads point to Vps10 localised to the vacuolar membrane in *Δsnx3*. Scale bar: 5 μm. (B) Quantification of data from A. Pearson correlation coefficients values were calculated using a Python-based app on FM4-64 masks, defining a zone close to the red signal around the vacuolar membrane. At least five confocal planes per experiment with 20–30 cells each from three independent experiments were analysed. Data are presented as mean values $+/-$ standard deviation. Asterisks indicate statistical significance evaluated by a two-tailed unpaired *t* test. ****$P < 0.0001$. See also Fig. EV7B for biological replicate variability. (C) Same experiment as in (A), but with cells expressing Ear1^mNG instead of Vps10^mNG. Scale bar: 5 μm. (D) Quantification of data from (C), performed as in (B). ****$P < 0.0001$. (E) Same experiment as in (A), but with cells carrying Ste13^mNG instead of Vps10^mNG. Scale bar: 5 μm. (F) Quantification of data from (E), performed as in (B). ****$P < 0.0001$. Source data are available online for this figure.

phospho-(1'-myo-inositol-3'-phosphate) (PI3P). 1,2-dioleoyl-sn-glycero-3-phosphoethanolamine-N-(Cyanine 5.5) (Cy5.5 PE); 1,2-dioleoyl-sn-glycero-3-[(N-(5-amino-1-carboxypentyl)iminodiacetic acid)succinyl] (nickel salt) (DGS-NTA(Ni)). All lipids were dissolved in chloroform and stored under argon at -20 °C and used within 2 months. Phosphatidylinositol phosphates were dissolved in chloroform/methanol/water (20:9:1). Texas Red DHPE (Thermofisher cat. T1395MP) was purchased as a mixed isomer. The para isomer was separated by thin-layer chromatography as previously described (Dar et al, 2015).

## Cell culture, strains and plasmids

SEY6210 yeast cells were grown at 30 °C in YPD (1% w/v yeast extract, 2% peptone, 2% dextrose) medium. Genes were deleted by replacing a complete open reading frame with a marker (Janke et al, 2004; Güldener et al, 1996) or CRISPR technique (Laughery et al, 2015) (see Tables 1–3 for a list of strains, plasmids and PCR primers used in this study). Gene tagging was performed as described (Sheff and Thorn, 2004) or with CRISPR (Laughery et al, 2015). Strains used for expression and purification of the Retromer complex have been described previously (Gopaldass et al, 2024).

## Live microscopy

Vacuoles were stained with FM4-64 essentially as described (Desfougères et al, 2016). An overnight preculture in SC medium was used to inoculate a 10 ml culture. Cells were then grown in SC at 30 °C and 150 rpm to an $OD_{600}$ between 0.6 and 1.0. The culture was diluted to an $OD_{600}$ of 0.4, and FM4-64 was added to a final concentration of 10 μM from a 10 mM stock in DMSO. Cells were labelled for 60 min with FM4-64, washed three times in fresh media, and then incubated for 60 min in media without FM4-64. Just before imaging, cells were concentrated by a brief low-speed centrifugation and placed on a glass microscopy slide overlaid with a 0.17-mm glass coverslip. Z-stacks were taken with a spacing of 0.3 μm and assembled into maximum projections. Imaging was performed with a NIKON Ti2 spinning disc confocal microscope with a 100×1.49 NA lens and two Photometrics Prime BSI cameras. Image analysis was performed with ImageJ as described in (Gopaldass et al, 2023) for SMTubes. Pearson's correlation coefficient was used to quantify the colocalization between Vps5, 17 and Snx3 using the "Just Another Colocalization Plug-in" (JACoP) in ImageJ on background-subtracted maximum projected images. A customised Python-based app was used to create masks defining a zone close to the FM4-64 signal around the vacuolar membrane and calculate Pearson's correlation coefficient of Vps10 and other cargoes on these FM4-64 masks. All performed experiments were repeated at least three times. SD and SEM calculation and plotting were done with PRISM GraphPad software.

## Protein immunoprecipitation and Western blotting

The interaction between SNX3, Retromer, and VPS5 was examined using a co-immunoprecipitation (co-IP) protocol adapted from (Suzuki et al, 2019). Yeast strains endogenously expressing Snx3-V5, Vps35-yomCherry, and Vps5-HA were inoculated in fresh YPD from saturated pre-cultures, grown overnight at 30 °C and harvested at mid-logarithmic phase ($OD_{600} = 0.8–1.0$). A total of 50 mL of culture was collected and washed twice with immuno-precipitation (IP) buffer (20 mM HEPES-KOH, pH 7.2, 0.2 M sorbitol, 50 mM potassium acetate, 1× protease inhibitor cocktail (200 μM pefablock, 11 μM leupeptin, 500 μM o-phenanthroline, 7 μM pepstatin A), 1 mM PMSF, followed by centrifugation to pellet the cells.

Cell lysis was performed using 0.5-mm zirconia beads in 300 μL IP buffer at 4 °C for 5 min at 2500 rpm on an IKA® VIBRAX® VXR orbital shaker. Subsequently, 300 μL of IP buffer containing either 1.0% Triton X-100 (for Vps35 immunoprecipitation) or 1.5% Triton X-100 (for Snx3 immunoprecipitation) was added to reach final detergent concentrations of 0.5% and 0.75%. Lysates were incubated on a rotating wheel at 4 °C for 2 h, followed by clarification through centrifugation at 500×*g* for 5 min and then at 17,500×*g* for 10 min. Protein concentrations in the supernatants were quantified using a NanoDrop spectrophotometer and the absorption at 280 and 205 nm. 10 mg of total protein was used for each immunoprecipitation reaction. Samples were incubated with pre-equilibrated anti-RFP magnetic particles (RFP-Trap Magnetic Particles, Chromotek) or anti-V5 magnetic agarose beads (V5-Trap Magnetic Agarose, Chromotek) for 2 h at 4 °C. After incubation, beads were washed three times with IP buffer containing the appropriate Triton X-100 concentration (0.5% or 0.75%). Prior to the final wash (performed with detergent-free IP buffer), the beads were transferred to fresh microcentrifuge tubes. Bound proteins were eluted by boiling the beads at 95 °C for 10 min in 4× NuPAGE sample buffer supplemented with 10 mM DTT, 2 mM PMSF, and 1× PIC. Protein samples were subsequently resolved by SDS-PAGE on freshly cast 4–18% polyacrylamide gels.

**Table 1.  Strains used in this study.**

| Strain | Genotype | Reference |
|---|---|---|
| CUY105 | *MATa his3Δ200 leu2Δ0 met15Δ0 trp1Δ63 ura3Δ0* | EUROSCARF |
| CUY100 | *MATalpha his3Δ200 leu2Δ0 lys2Δ0 met15Δ0 trp1Δ63 ura3Δ0* | EUROSCARF |
| CUY9228 | *CUY105, VPS5pr::natNT1-GAL1pr VPS17pr::kanMX-GAL1pr VPS5::TAP-URA3 vps35::HIS3* | Purushothaman et al, 2017 |
| CUY9495 | *CUY100, VPS26pr::HIS3-GAL1pr VPS29pr::natNT2-GAL1pr VPS35pr::hphNT1-GAL1 VPS35::TAP-kanMX vps5::TRP1 vps17::LEU2* | Purushothaman et al, 2017 |
| CUY9711 | *CUY105, VPS5pr::natNT1-GAL1pr VPS17pr::kanMX-GAL1pr VPS5::TAP-URA3 vps35::HIS3 VPS17::GFP-hphNT1* | Purushothaman et al, 2017 |
| CUY9932 | *CUY105, VPS26pr::HIS3-GAL1pr VPS29pr::natNT2-GAL1pr VPS35pr::hphNT1-GAL1 VPS29-mClover::kan VPS26::TAP-URA3 vps5Δ::TRP1* | Purushothaman et al, 2017 |
| CUY9935 | *CUY105, VPS26pr::HIS3-GAL1pr VPS29pr::natNT2-GAL1pr VPS35pr::hphNT1-GAL1pr VPS29-mCherry::kan VPS26::TAP-URA3 vps5Δ::TRP1* | Purushothaman et al, 2017 |
| SEY6210 | *MATα leu2-3,112 ura3-52 his3-Δ200 trp1-Δ901 suc2-Δ9 lys2-801; GAL* | Scott Emr |
| SEY6211 | *MATa leu2-3,112 ura3-52 his3-Δ200 trp1-Δ901 ade2-101 suc2-Δ9; GAL* | Scott Emr |
| CA369 | *SEY6210 Ear1-mNG (URA), vps35Δ, LEU::pRS405 Vps35prom-Vps35-mCherry-ADHterm, Vps5-3HA (TRP)* | This study |
| CA370 | *SEY6210 Ear1-mNG (URA), Snx3-V5, Vps5-3HA (TRP)* | This study |
| CA371 | *SEY6210 Ear1-mNG (URA), vps35Δ, LEU::pRS405 Vps35prom-Vps35-mCherry-ADHterm, Snx3-V5, Vps5-3HA (TRP)* | This study |
| CA372 | *SEY6210 Ear1-mNG (URA), vps35Δ, LEU::pRS405 Vps35prom-Q181G182R185E186-Vps35-mCherry-ADHterm, Snx3-V5, Vps5-3HA (TRP)* | This study |
| CA310 | *SEY6210 Ear1-mNG (URA), vps35Δ* | This study |
| CA311 | *SEY6210 Ear1-mNG (URA), vps35Δ, LEU::pRS405 Vps35prom-Vps35-mCherry-ADHterm* | This study |
| CA312 | *SEY6210 Ear1-mNG (URA), vps35Δ, LEU::pRS405 Vps35prom-Q181G182R185E186-Vps35-mCherry-ADHterm* | This study |
| CA318 | *SEY6210 Ear1-mNG (URA), Snx3-V5* | This study |
| CA319 | *SEY6210 Ear1-mNG (URA), vps35Δ, LEU::pRS405 Vps35prom-Vps35-mCherry-ADHterm, Snx3-V5* | This study |
| CA320 | *SEY6210 Ear1-mNG (URA), vps35Δ, LEU::pRS405 Vps35prom-Q181G182R185E186-Vps35-mCherry-ADHterm, Snx3-V5* | This study |
| CA321 | *SEY6210 Ear1-mNG (URA), vps35Δ, LEU::pRS405 Vps35prom-Vps35-mCherry-ADHterm, Snx3-V5, vps17Δ* | This study |
| CA347 | *SEY6210 Ear1-mNG (URA), vps35Δ, LEU::pRS405 Vps35prom-Vps35-mCherry-ADHterm, snx3Δ* | This study |
| CA307 | *SEY6210 Vps10-mNG, vps35Δ, URA::pRS406 Vps35prom-Vps35-mCherry-ADHterm* | This study |
| CA308 | *SEY6210 Vps10-mNG, vps35Δ, URA::pRS406 Vps35prom-Vps35-Q181G182R185E186-mCherry-ADHterm* | This study |
| CA347 | *SEY6210 Vps10-mNG, vps35Δ, URA::pRS406 Vps35prom-Vps35-mCherry-ADHterm, snx3Δ* | This study |
| NG450 | *SEY6210 snx3Δ::kanMX4* | This study |
| AM4134 | *SEY6211 vps17Δ::natNT2* | This study |
| SRC10 | *SEY6210 Δvps5 Vam10-V5::LEU2* | This study |
| SRC37 | *SEY6210 Δvps5 Vam10-V5::LEU2 Vps10-mNeonGreen* | This study |
| SRC39 | *SEY6210 Δvps5 Vam10-V5::LEU2 Ear1-mNeonGreen* | This study |
| SRC41 | *SEY6210 Δvps5 Vam10-V5::LEU2 mNeonGreen-Ste13* | This study |
| SRC40 | *SEY6210 Δvps5 Vam10-V5::LEU2 Kex2-mNeonGreen* | This study |
| SRC38 | *SEY6210 Δvps5 Vam10-V5::LEU2 mNeonGreen-Pep12* | This study |
| SRC42 | *SEY6211 vps17Δ::natNT2 Vps10-mNeonGreen* | This study |
| SRC44 | *SEY6211 vps17Δ::natNT2 Ear1-mNeonGreen* | This study |
| SRC46 | *SEY6211 vps17Δ::natNT2 mNeonGreen-Ste13* | This study |
| SRC45 | *SEY6211 vps17Δ::natNT2 Kex2-mNeonGreen* | This study |
| SRC43 | *SEY6211 vps17Δ::natNT2 mNeonGreen-Pep12* | This study |
| SRC62 | *SEY6210 snx3Δ::kanMX4 Vps10-mNeonGreen* | This study |
| SRC64 | *SEY6210 snx3Δ::kanMX4 Ear1-mNeonGreen* | This study |
| SRC66 | *SEY6210 snx3Δ::kanMX4 mNeonGreen-Ste13* | This study |
| SRC63 | *SEY6210 snx3Δ::kanMX4 Kex2-mNeonGreen* | This study |
| SRC65 | *SEY6210 snx3Δ::kanMX4 mNeonGreen-Pep12* | This study |
| SRC30 | *SEY6210 Vps5-yomCherry ::SpHIS5 Vps17-mNeonGreen ::CaURA3* | This study |
| SRC92 | *SEY6210 Vps5-yomCherry ::SpHIS5 Vps17-mNeonGreen ::CaURA3, vps35Δ* | This study |

**Table 1.** (continued)

| Strain | Genotype | Reference |
|--------|----------|-----------|
| SRC31 | *SEY6210 Vps5-yomCherry ::SpHIS5 Snx3-mNeonGreen ::CaURA3* | This study |
| SRC93 | *SEY6210 Vps5-yomCherry ::SpHIS5 Snx3-mNeonGreen ::CaURA3, vps35Δ* | This study |
| SRC22 | *SEY6210 Vps10-mNeonGreen* | This study |
| SRC23 | *SEY6210 Ear1-mNeonGreen* | This study |
| SRC26 | *SEY6210 mNeonGreen-Ste13* | This study |
| SRC24 | *SEY6210 Kex2-mNeonGreen* | This study |
| SRC25 | *SEY6210 mNeonGreen-Pep12* | This study |

**Table 2.** Plasmids used in this study.

| Number | Name |
|--------|------|
| CA171 | pRS406 Vps35prom-Vps35-mCherry-ADHterm |
| CA173 | pRS406 Vps35prom-Vps35-Q181G182R185E186-mCherry-ADHterm |
| CA174 | pRS405 Vps35prom-Q181G182R185E186-Vps35-mCherry-ADHterm |
| CA175 | pRS405 Vps35prom-Vps35-mCherry-ADHterm |

## Protein purification

TAP-tagged Retromer complex was extracted from yeast as previously described (Purushothaman et al, 2017; Purushothaman and Ungermann, 2018). Briefly, a 50 mL preculture of cells was grown for 24 h in YPGal medium. The next day, two 1 L cultures in YPGal were inoculated with 15 mL of preculture and grown for 20 h at 30 °C and 150 rpm to late log phase (OD$_{600}$ = 2–3). All the following steps were performed at 4 °C. Cells were pelleted and washed with 1 pellet volume of cold RP buffer (Retromer Purification buffer: 50 mM Tris pH 8.0, 300 mM NaCl, 1 mM MgCl$_2$, 1 mM PMSF, and home-made protein inhibitor cocktail (200 μM pefablock, 11 μM leupeptin, 500 μM o-phenanthroline, 7 μM pepstatin A). Pellets were either processed immediately or flash-frozen in liquid nitrogen and stored at -80 °C. For cell lysis, the pellet was resuspended in one volume of RP buffer and passed once through a French press (One shot cell disruptor, Constant Systems LTD, Daventry, UK) at 2.2 Kpsi. DNase I was added to the lysate (final concentration 0.1 mg/mL) followed by a 20 min incubation on a rotating wheel. The lysate was precleared by centrifugation for 30 min at 45,000×g in a Beckman JLA 25.50 rotor and cleared by a 60 min centrifugation at 150,000×g in a Beckman Ti 60 rotor. The cleared supernatant was passed through a 0.2 μm filter and transferred to a 50 mL Falcon tube. In all, 1 mL IgG bead suspension (GE Healthcare, cat 17-0969-01) was added to the supernatant. After 60 min incubation on a rotating wheel, beads were spun down and washed three times with RP buffer and resuspended in 2 mL RP buffer. 250 μg of home-purified HIS-TEV protease from *E. coli* was added to the beads. After 30 min incubation at 4 °C, beads were centrifuged, the supernatant containing purified Retromer subcomplex was collected and concentrated on a 100 kDa cutoff column (Pierce™ Protein Concentrator PES, 100 K MWCO). The concentrated protein fraction was re-diluted in RP buffer and reconcentrated 3 times. This final step allowed for the removal of TEV protease and a high enrichment of intact complexes.

Snx3 (with or without a fluorescent tag (GFP or mScarletI) at the N-terminus) was cloned into the pGE-6P vector modified to contain a HRV3C protease cleavage site after the GST. Plasmids were transformed into *E. coli* BL21. A 50 mL preculture was used to inoculate 2 litres of LB media (37 °C). Cells were grown to an OD$_{600}$ of 0.8–0.9. The cultures were then cooled to 16 °C on ice, and IPTG (Roche) was added to a final concentration of 0.5 mM. Cells were incubated overnight at 16 °C, pelleted, washed with ice-cold RP buffer and resuspended in one pellet volume of RP buffer + 1 mM DTT. Cells were lysed in a French Press (One Shot cell disruptor, Constant Systems LTD, Daventry, UK). Insoluble material was pelleted at 30,000×g for 30 min in a Beckman JA25.50 rotor at 4 °C. Supernatants were filtered through a 0.2-μm filter (Sarstedt, Germany), and lysates were passed three times over 1 ml of Glutathion-Sepharose (Qiagen). The resin was washed with 30 ml of RP buffer, and resuspended in cleavage buffer (50 mM Tris pH = 8.0, 300 mM NaCl, 1 mM EDTA, 1 mM DTT), and 150 μg of HRV3C protease was added. Beads were incubated overnight at 4 °C. The supernatant containing purified Snx3 was analysed by SDS gel electrophoresis. For $^{GFP}$Snx3, the protein was further purified by FPLC to remove unlabelled Snx3.

Proteins were concentrated to ~2 mg/mL, aliquoted in 10 μL fractions and flash-frozen in liquid nitrogen. Proteins were stored at -80 °C and used within 3 months. Thawed aliquots were used only once.

## Peptide synthesis and labelling

Peptides were synthesised at the Protein and Peptide Chemistry Facility, University of Lausanne (Switzerland) and analysed by mass spectrometry. The homogeneity of all peptides was >90%, as indicated by analytical HPLC. For fluorescent labelling, 3–4 mg of peptides were dissolved in 100 μL DMSO and incubated at 20 °C for 12 h with an equimolar amount of maleimide-coupled fluorophore (Alexa Fluor™ 488 or 546 C5 Maleimide, Invitrogen). Labelled peptides were then purified by HPLC.

**Cargo peptide sequences**: numbering indicates the amino acid position in the full-length protein. The sequence in brackets at the end was added to permit labelling with a maleimide-coupled fluorophore.

Ste13: HHHHHHGGGG-$^{78}$MRPRRESFQFNDIENQH
Ear1: HHHHHHGGGG-$^{475}$GKKIINEEINLDSL-(GGC)

**Table 3. Primers used in this study.**

| Primer name | Sequence (5′–3′) |
|---|---|
| CA367 Fw Vps10 ctrl C term | GCTGGCCACGATGAAGAC |
| CA368 Rv Vps10 ctrl C term | GATATTCTGTCTAAAGGATCTGCTCG |
| RC99_Ear1-mNeonGreen(A2739)-F | TTCAGAATTTGATGATTACGAAAGCAGGATGCATGGCATAggtagtgctggaagtgcag |
| RC100_Ear1-mNeonGreen(A2739)-R | GGGCTAGTGTTTCAGCCTTACTATCTCATGCATTTTCGTAgcaggttaacctggcttatcg |
| RC103_mNeonGreen-Ste13(A2743)-F (for CRISPR) | GTTAGGGTGTTATTCGTGTAAAAAATATAGAAAGCCCCTAatggtctctaagggtgaagaag |
| RC104_mNeonGreen-Ste13(A2743)-R (for CRISPR) | GACTATTCTTCCTTTTATGCGAATGAGTTGAAGCAGACATagctgcacttccagcacta |
| RC127_Snx3mNeonGreen_F | AGTTCTCGTGAGGTTCATTGAAGCTGAAAAGTTTGTCGGCggtagtgctggaagtgca |
| RC128_Snx3mNeonGreen_R | TATATAATCTATATTATTTATTCACGTAAAAGAGTTCTTTggatggcggcgttagtatc |
| oTC52_Vps5yomCherry-F | ATGCATCGAGCTTTGGGAGACATTCTACCAAACCAATCTTggtgacggtgctggtttta |
| oTC53_Vps5yomCherry-R | AGGAACGTGACACATAAAGTTATTGTATACAGATCATCTAtcgatgaattcgagctcg |
| RC105_mNeonGreen-Pep12(A2743)-F (for CRISPR) | AGAAGAATATAACGTAAATTACTACAATAATTGTGTTGAGatggtctctaagggtgaagaag |
| RC106_mNeonGreen-Pep12(A2743)-R (for CRISPR) | CCAAACGGCTTCATTATCACCACCAAAAAATTCGTCTTCtGACATagctgcacttccagcacta |
| RC101_Kex2-mNeonGreen(A2739)-F | AGAATTACAGCCTGATGTTCCTCCATCTTCCGGACGATCGggtagtgctggaagtgcag |
| RC102_Kex2-mNeonGreen(A2739)-R | AATGCTATTTTGTAATTTGAAGCTTTCTGTACATATCGAAgcaggttaacctggcttatcg |
| RC107_Pep12(N)-g24-F | CAATAATTGTGTTGAGATGTGTTTTAGAG |
| RC108_Pep12(N)-g24-R | CTAGCTCTAAAACACATCTCAACACAATTATTGACGT |
| RC109_Ste13(N)-g35-F | TGAGTTGAAGCAGACATTAGGTTTTAGAG |
| RC110_Ste13(N)-g35-R | CTAGCTCTAAAACCTAATGTCTGCTTCAACTCAACGT |
| CA374 Fw Vps35 del cassette | TAA AAG GAG GAG GAC GAG AAA GAA GAA GCT GAA AAA CAC ATT AAA TAA CGT ACT TGC TCT TCG TAC ATG CCC AAG ATA AA |
| CA375 Rv Vps35 del cassette | TTT ATC TTG GGC ATG TAC GAA GAG CAA GTA CGT TAT TTA ATG TGT TTT CA GCT TCT TCT TTC TCG TCC TCC TCC TTT TA |

Vps10: HHHHHHGGGGGG-[1423]GGFARFGEIRLGDDGLIE-(GGC).

## Supported membrane tubes

SMTubes were generated as previously described (Dar et al, 2015; Gopaldass et al, 2024). Briefly, glass coverslips were first washed with 3 M NaOH for 5 min and rinsed with water before a 60 min treatment with piranha solution (95% $H_2SO_4$/30% $H_2O_2$ 3:2 v/v). Coverslips were rinsed with water and dried on a heat block at 90 °C. Coverslips were then silanized with 3-glycidyloxypropyltrimethoxysilane (Catalogue no. 440167, Sigma) for 5 h under vacuum, rinsed with acetone and dried. Polyethylene glycol coating was performed by placing the coverslips in a beaker containing PEG400 (Sigma) at 90 °C for 60 h. Coverslips were washed with distilled water and stored for up to 2 months at room temperature in a closed container.

To generate supported membrane tubes, lipids were mixed from 10 mg/mL stocks in a glass vial and diluted to a final concentration of 1 mg/mL in chloroform. The same lipid mix was used throughout this study (5% PI3P, 15% DOPS, 0.1% Texas Red DHPE, 79.5% egg-PC). Lipids were then spotted (typically 1 µL, corresponding to about 1 nmol) on the coverslips and dried for 30 min under vacuum. The coverslip was mounted on an IBIDI 6-channel µ-slide (µ-Slide VI 0.4. IBIDI, Cat. No: 80606). Lipids were hydrated for 15 min with buffer (PBS pH 7.2: 137 mM NaCl,

2.7 mM KCl, 10 mM $Na_2HPO_4$, 1.8 mM $KH_2PO_4$)) and SMTs were generated by injecting PBS into the chamber using an Aladdin Single-Syringe Pump (World Precision Instruments, model AL-1000) at a flow rate of 1.5 mL/min for 5 min. SMTs were left to stabilise without flow for 5 min before the start of the experiment. Protein stocks (typically 1–2 µM) were first diluted in PBS and then injected into the chamber at a flow rate of 80 µL per minute. Tubes were imaged with a NIKON Ti2 spinning disc confocal microscope equipped with a 100×1.49 NA objective.

## Quantification of SMT fluorescence

SMT fluorescence was quantified with ImageJ as described previously (Gopaldass et al, 2023). Briefly, line scan analysis was performed along tubules using ImageJ. For each line scan, a Gaussian curve was fitted, and the maximum height was extracted. Maximum height was then plotted against the tube length for all channels. For quantification of the radius of the tubes, lipid fluorescence values of a tubule underneath a constricted protein domain, extracted from the series of line scans described above, were sorted in ascending order. The curve typically showed two plateaus, the lower corresponding to the constricted state and the higher to the non-constricted one. Plotting the corresponding FP values confirmed that the FP-labelled protein localised to the constricted zone. For each tube, the zones corresponding to the

constricted and non-constricted areas were determined manually, and the mean fluorescence value was used to calculate the tube radius or the FP intensity in the corresponding region.

## Data availability

This study includes no data deposited in external repositories.

The source data of this paper are collected in the following database record: biostudies:S-SCDT-10_1038-S44318-026-00716-0.

## Peer review information

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

## Acknowledgements

We thank Arianna Ravera from the DCSR at CI-UNIL for implementing the Python-based app used for image analysis. This work has been supported by grants from the Swiss National Science Foundation to AM (179306, 204713 and 10.006.083), and by EMBO to SRC (ALTF 240-2023).

## Author contributions

**Navin Gopaldass**: Conceptualisation; Data curation; Formal analysis; Validation; Investigation; Methodology; Writing—original draft; Writing—review and editing. **Sudeshna Roy Chowdhury**: Data curation; Investigation; Methodology; Writing—original draft; Writing—review and editing. **Ana Catarina Alves**: Data curation; Investigation; Methodology; Writing—original draft; Writing—review and editing. **Lydie Michaillat Mayer**: Resources; Investigation; Methodology. **Véronique Comte-Miserez**: Resources; Investigation; Methodology. **Andreas Mayer**: Conceptualisation; Resources; Supervision; Methodology; Writing—original draft; Project administration; Writing—review and editing.

Source data underlying figure panels in this paper may have individual authorship assigned. Where available, figure panel/source data authorship is listed in the following database record: biostudies:S-SCDT-10_1038-S44318-026-00716-0.

## Disclosure and competing interests statement

The authors declare no competing interests.

# Expanded View Figures

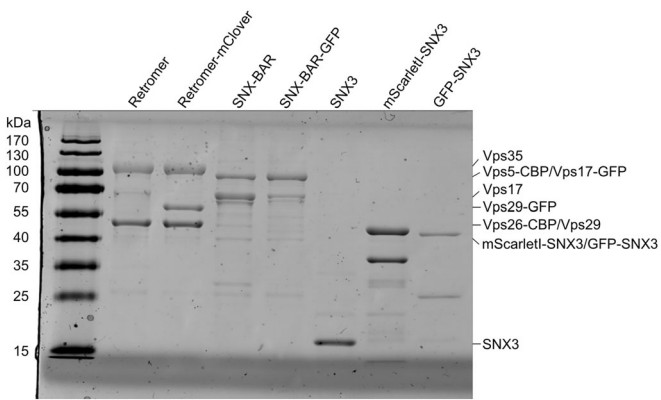

**Figure EV1.  Protein preparations.**

Coomassie-stained SDS-PAGE gel of the protein preparations used in the in vitro experiments.

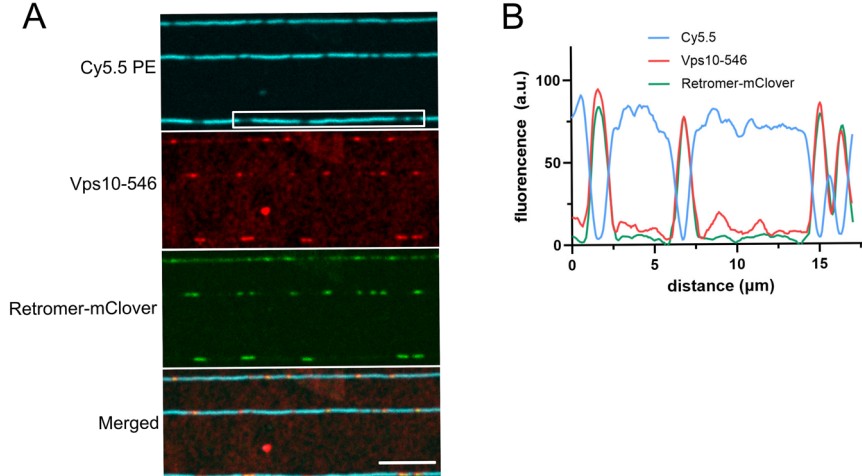

**Figure EV2.   Vps10 cargo peptide concentration by SNX-BAR-Retromer coats.**

(**A**) SMTubes were formed and pre-incubated with 5 µM fluorescently labelled cargo peptide Vps10-546 for 10 mins, followed by addition of 25 nM SNX-BAR and 25 nM Retromer^mClover until coat formation became apparent (2–3 min). After a brief wash with phosphate-buffered saline (PBS), tubes were imaged by spinning disc confocal microscopy. Scale bar: 2 µm. (**B**) Line scan of the region boxed in A.

A

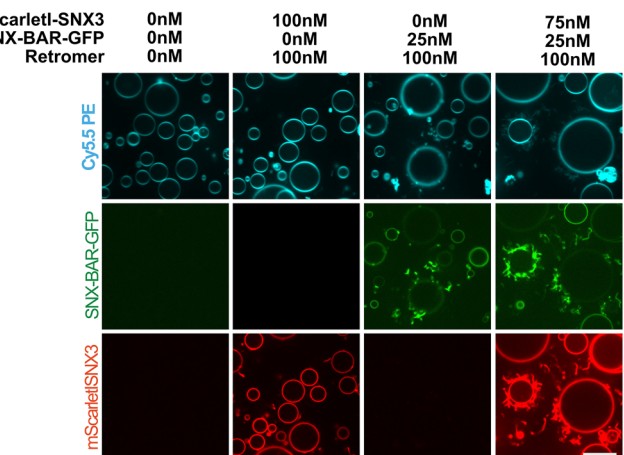

B

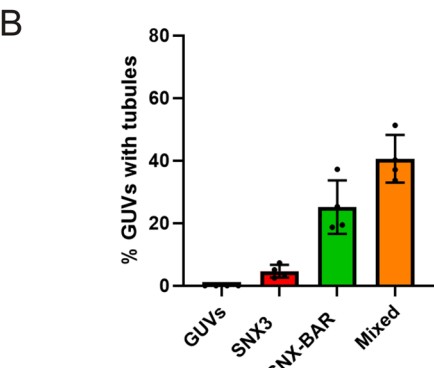

**Figure EV3.** **Tubulation of GUVs by hybrid coats in the presence of SNX-BAR and Snx3 cargos.**

(A) GUVs labelled with Cy5.5-PE were incubated with the indicated concentrations of Retromer, mScarletISnx3 and SNX-BARGFP, and with 5 μM each of Ear1 and Vps10 peptide. After 1 h of incubation, the GUVs were imaged by confocal microscopy. (B) Quantification of the percentage of GUVs with tubules shown in (A). The mean and standard deviation from 4 independent experiments were determined. Total number of scored items: GUVs only ($n = 108$); Snx3/Retromer ($n = 14$); SNX-BAR/Retromer ($n = 190$); Snx3/SNX-BAR/Retromer ($n = 167$). Scale bar: 5 μm.

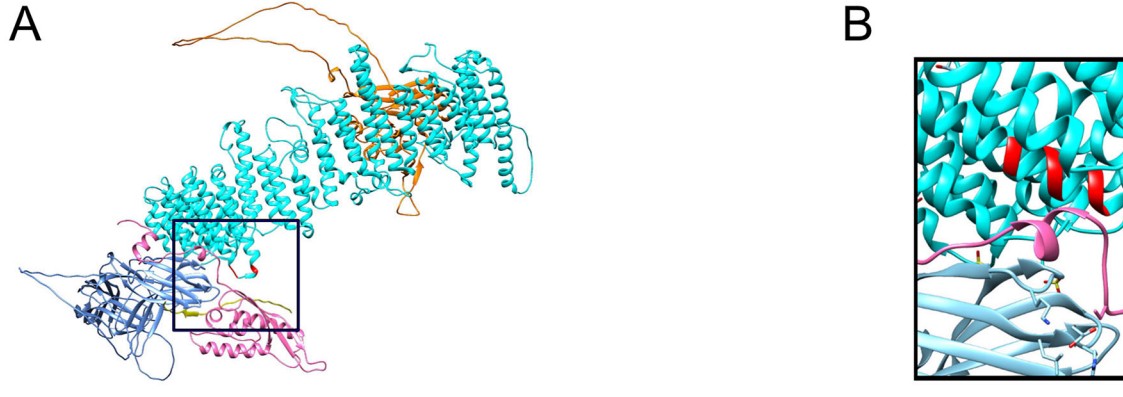

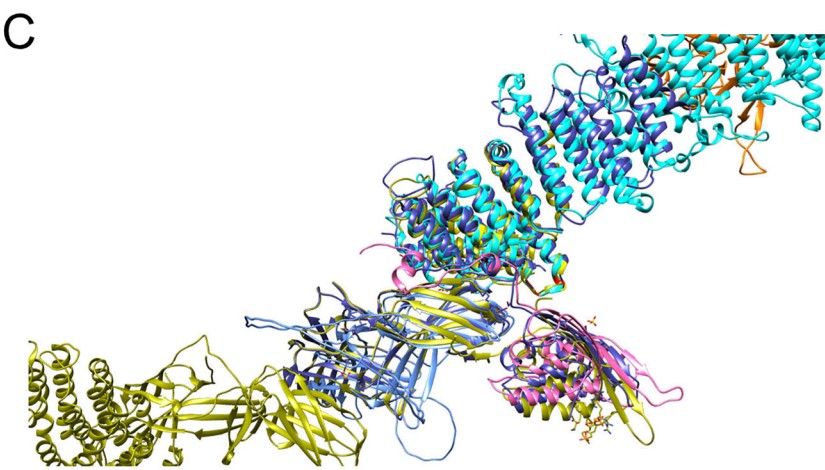

**Figure EV4. Structural context of the Vps35^{QGRE} mutant.**

(A) Alphafold 3 model of the S. cerevisiae Retromer in the presence of the Ear1 cargo peptide and Snx3. Snx3: pink; Vps26: blue; Vps35: cyan; Ear1 peptide: yellow; Substituted QGRE residues: red (B). Region of the human Retromer-Snx3 crystal structure (PDB 5F0M) from Hierro et al that corresponds to the boxed region of the Alphafold model in A The residues predicted by the PDB-PISA software (https://www.ebi.ac.uk/pdbe/pisa/) to participate in Vps35-Snx3 interface are shown in red. The circle highlights the substituted residues in Vps35^{QGRE} (C). Overlay of the Retromer-Snx3 structures from (H). sapiens (PDB 5F0M; purple) and T. thermophila (PDB 7BLQ) with the Alphafold model of S. cerevisiae (pink: Snx3, cyan: Vps35, blue: Vps26). (D) Alignment of the Vps35 sequences from *H. sapiens*, *T. thermophila* and *S. cerevisiae*. Residues substituted in Vps35^{QGRE} are boxed in red. The interface predicted by PISA is marked in bold type.

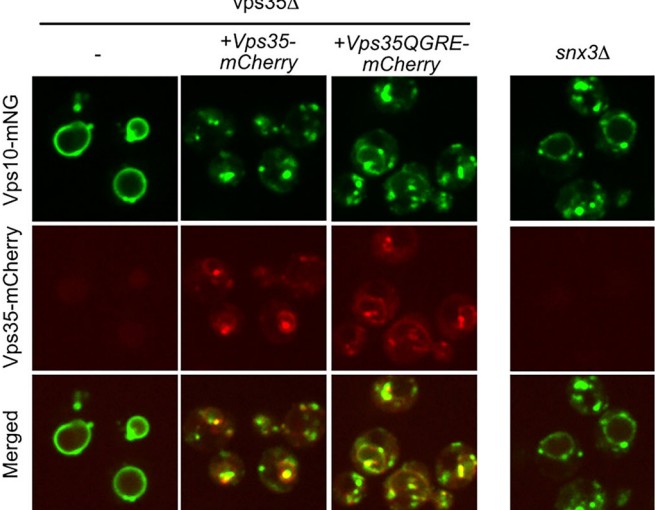

**Figure EV5.  Mislocalization of Vps10 in *vps35^QGRE* cells.**

Logarithmically growing *vps35Δ* cells expressing genomically tagged Vps10^mNG were transformed with integrative plasmids expressing Vps35^mCherry (WT), Vps35^QGRE-mCherry, or nothing. The cells were logarithmically grown overnight and analysed by spinning disc microscopy. Scale bar: 5 μm.

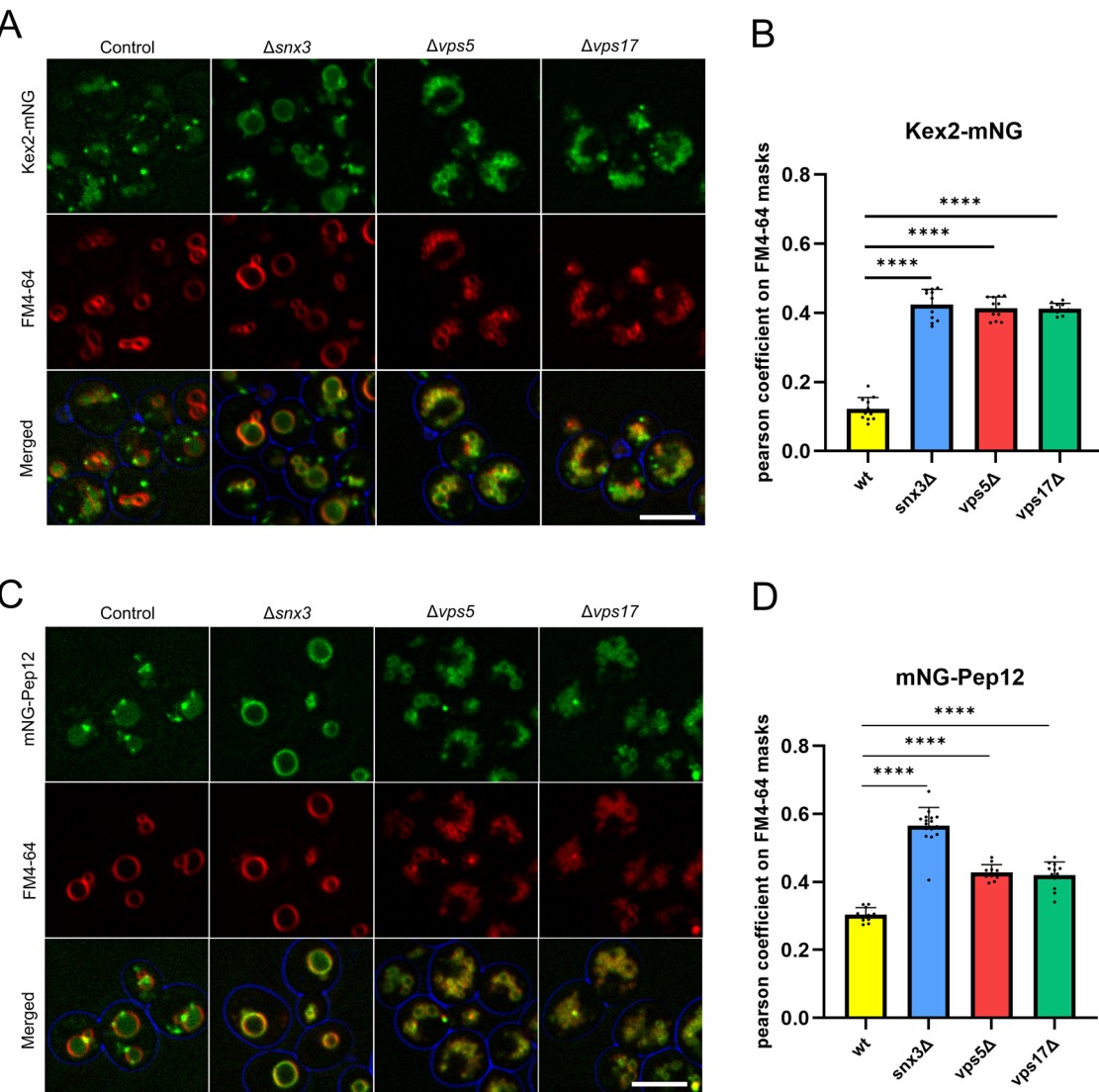

**Figure EV6. Mutual dependence of Snx3 and SNX-BARs for cargo recycling.**

(A) Kex2 distribution. Logarithmically growing cells expressing Kex2^mNG in wild-type, Snx3Δ, vps5Δ, and vps17Δ background were stained with FM4-64 and analysed by confocal microscopy. Average intensity projections of confocal z-stacks taken at a z-interval of 0.3 μm are shown. (B) Quantification of data from (A). Pearson correlation coefficients values were calculated using a Python-based app on FM4-64 masks, defining a zone close to the red signal around the vacuolar membrane. At least 5 confocal planes per experiment with 20–30 cells each from two independent experiments were analysed. Data are presented as mean values +/− standard deviation. Asterisks indicate statistical significance using two-tailed unpaired $t$ tests. ****$P < 0.0001$. See also Fig. EV7C for biological replicate variability. (C) Same experiment as in (A), but with cells expressing ^mNGPep12 instead of Kex2^mNG. (D) Quantification of data from (C), performed as in (B). Asterisks indicate statistical significance using two-tailed unpaired $t$ tests. ****$P < 0.0001$.

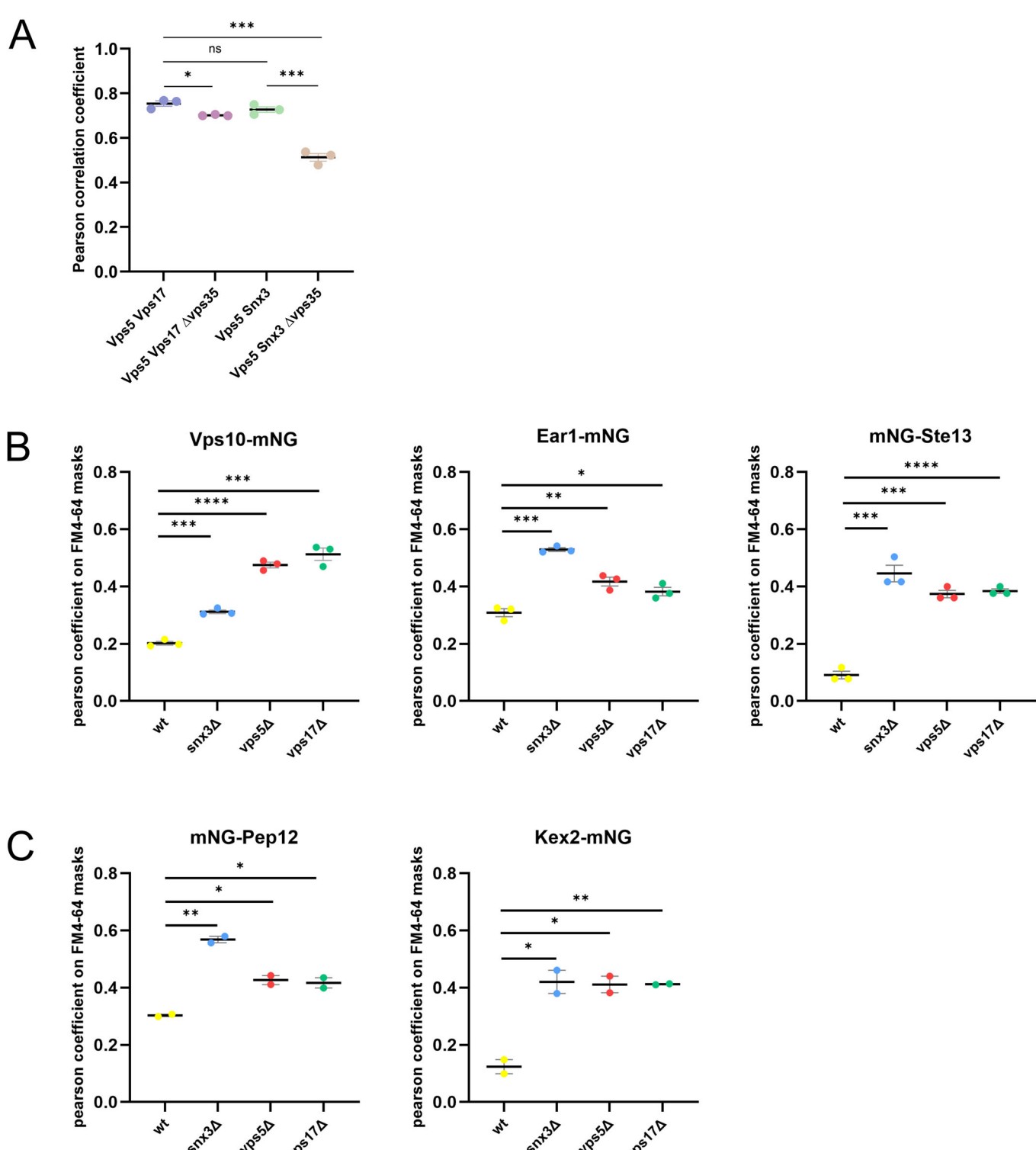

**Figure EV7. Experiment-to-experiment variation in microscopic in vivo analyses.**

(**A**) The quantification data from Fig. 7B was pooled per biological replicate to visualise experiment to experiment variation. (**B**) The quantification data from Fig. 8B,D,F was pooled per biological replicate to visualise experiment to experiment variation. (**C**) The quantification data from Fig. EV6B,D was pooled per biological replicate to visualise experiment to experiment variation. For all graphs in this figure bars represent the mean values and standard error mean. Two-tailed unpaired *t* tests were used to evaluate significance. ****$P < 0.0001$; ***$P < 0.001$; **$P < 0.01$ and *$P < 0.1$.

