## [Peer Review File · The EMBO Journal]

Hybrid endosomal coats contain different classes of sorting nexins

Navin Gopaldass, Sudeshna Chowdhury, Ana Alves, Lydie Michailat Mayer, Véronique Comte-Miserez, and Andreas Mayer

Corresponding author(s): Andreas Mayer (andreas.mayer@unil.ch) , Navin Gopaldass (Navin.Gopaldass@unil.ch)

Review Timeline:

Submission Date:	28th Jul 25
Editorial Decision:	8th Sep 25
Revision Received:	24th Nov 25
Editorial Decision:	23rd Dec 25
Revision Received:	14th Jan 26
Accepted:	23rd Jan 26

Editor: William Teale

Transaction Report:

Dear Andreas,

Thank you again for the submission of your manuscript entitled "Hybrid endosomal coats containing different classes of sorting nexins" and for your patience during the review process. We have now received reports from two referees, which I copy below.

As you can see from their comments, both thought that your manuscript could represent a timely and well-directed study. Both also point out some issues that will require your attention before your manuscript can be published in The EMBO Journal. I am particularly concerned about Referee #1's enquiring about whether, and within a realistic time-frame, the in vivo data you present could be strengthened.

Based on the overall interest expressed in the reports, however, I would like to invite you to address the comments of all referees in a revised version of the manuscript. I should add that it is The EMBO Journal policy to allow only a single major round of revision and that it is therefore important to resolve the main concerns at this stage. I believe the concerns of the referees are reasonable and addressable, but please contact me if you have any questions, need further input on the referee comments or if you anticipate any problems in addressing any of their points. I am always available on Zoom should you wish to discuss the referees' reports or your responses to them. Please, follow the instructions below when preparing your manuscript for resubmission.

I would also like to point out that as a matter of policy, competing manuscripts published during this period will not be taken into consideration in our assessment of the novelty presented by your study ("scooping" protection). We have extended this 'scooping protection policy' beyond the usual 3 month revision timeline to cover the period required for a full revision to address the essential experimental issues. Please contact me if you see a paper with related content published elsewhere to discuss the appropriate course of action.

Again, please contact me at any time during revision if you need any help or have further questions.

Thank you very much again for the opportunity to consider your work for publication. I look forward to your revision.

Best regards,

William

William Teale, Ph.D.
Editor
The EMBO Journal

When submitting your revised manuscript, please carefully review the instructions below and include the following items:

- 1) a .docx formatted version of the manuscript text (including legends for main figures, EV figures and tables). Please make sure that the changes are highlighted to be clearly visible.
- 2) individual production quality figure files as .eps, .tif, .jpg (one file per figure).
- 3) a .docx formatted letter INCLUDING the reviewers' reports and your detailed point-by-point response to their comments. As part of the EMBO Press transparent editorial process, the point-by-point response is part of the Review Process File (RPF), which will be published alongside your paper.
- 4) a complete author checklist, which you can download from our author guidelines ([https://wol-prod-cdn.literatumonline.com/pb-assets/embo-site/Author Checklist%20-%20EMBO%20J-1561436015657.xlsx](https://wol-prod-cdn.literatumonline.com/pb-assets/embo-site/Author%20Checklist%20-%20EMBO%20J-1561436015657.xlsx)). Please insert information in the checklist that is also reflected in the manuscript. The completed author checklist will also be part of the RPF.
- 5) Please note that all corresponding authors are required to supply an ORCID ID for their name upon submission of a revised manuscript.
- 6) We require a 'Data Availability' section after the Materials and Methods. Before submitting your revision, primary datasets produced in this study need to be deposited in an appropriate public database, and the accession numbers and database listed

under 'Data Availability'. Please remember to provide a reviewer password if the datasets are not yet public (see <https://www.embopress.org/page/journal/14602075/authorguide#datadeposition>). If no data deposition in external databases is needed for this paper, please then state in this section: This study includes no data deposited in external repositories. Note that the Data Availability Section is restricted to new primary data that are part of this study.

Note - All links should resolve to a page where the data can be accessed.

8) For data quantification: please specify the name of the statistical test used to generate error bars and P values, the number (n) of independent experiments (specify technical or biological replicates) underlying each data point and the test used to calculate p-values in each figure legend. The figure legends should contain a basic description of n, P and the test applied. Graphs must include a description of the bars and the error bars (s.d., s.e.m.).

9) We would also encourage you to include the source data for figure panels that show essential data. Numerical data can be provided as individual .xls or .csv files (including a tab describing the data). For 'blots' or microscopy, uncropped images should be submitted (using a zip archive or a single pdf per main figure if multiple images need to be supplied for one panel). Additional information on source data and instruction on how to label the files are available at .

10) We replaced Supplementary Information with Expanded View (EV) Figures and Tables that are collapsible/expandable online (see examples in <https://www.embopress.org/doi/10.15252/embj.201695874>). A maximum of 5 EV Figures can be typeset. EV Figures should be cited as 'Figure EV1, Figure EV2" etc. in the text and their respective legends should be included in the main text after the legends of regular figures.

12) Our journal encourages inclusion of *data citations in the reference list* to directly cite datasets that were re-used and obtained from public databases. Data citations in the article text are distinct from normal bibliographical citations and should directly link to the database records from which the data can be accessed. In the main text, data citations are formatted as follows: "Data ref: Smith et al, 2001" or "Data ref: NCBI Sequence Read Archive PRJNA342805, 2017". In the Reference list, data citations must be labeled with "[DATASET]". A data reference must provide the database name, accession number/identifiers and a resolvable link to the landing page from which the data can be accessed at the end of the reference. Further instructions are available at .

13) In order to increase the reproducibility and reach of your work, The EMBO Journal includes a table of reagents that were used in the study. Please provide this along with your revisions.

Further instructions for preparing your revised manuscript:

- a point-by-point response to the referees' comments, with a detailed description of the changes made (as a word file).
- a word file of the manuscript text.

- individual production quality figure files (one file per figure)

- a complete author checklist, which you can download from our author guidelines

(<https://www.embopress.org/page/journal/14602075/authorguide>).

- Expanded View files (replacing Supplementary Information)

- a Reagents and Tools Table as part of the Methods section, which can be downloaded from our author guidelines

(<https://www.embopress.org/page/journal/14602075/authorguide#structuredmethods>)

We realize that it is difficult to revise to a specific deadline. In the interest of protecting the conceptual advance provided by the work, we recommend a revision within 3 months (7th Dec 2025). Please discuss the revision progress ahead of this time with the editor if you require more time to complete the revisions. Use the link below to submit your revision:

Referee #1:

Gopaldass et al present a set of data addressing one of the fundamental open questions in endosome biology- how do the litany of retrograde trafficking complexes and retromer flavours organise themselves? Using two complexes, the SNX-BAR:retromer and SNX3:retromer from yeast, they use an elegant in vitro system to make a number of important and interesting observations. The most interesting of which is that in the situation where both complexes are co-incubated on membranes, they co-localise to the same membrane patch, and moreover, there appears to be a constriction-property benefit from co-ordination of the two complexes. I find this finding extremely compelling, and it really challenges the retromer flavour 'handover' model that has become the accepted narrative in the last ten years. I have some issues with the manuscript and particularly the in vivo section, which I will detail below; however, overall, I find these experiments and the findings compelling and important.

I will start off by discussing the in vivo data, which I think is one of the weaker sections of what is an overall very well-performed and interoperated set of data and manuscript. Simply put, there is nothing presented to my mind which actually directly supports the in vitro data. I will list my criticisms of each of the supporting experiments. The colocalisation microscopy is not surprising that they colocalise somewhat; it is diffraction-limited imaging in a small model organism, regardless of the actual situation, they would be on the same endosome/endosomal subdomain. Are they looking at carriers here? Is there any evidence of this? As an aside, there is no negative control for the Pearsons, but this is a minor issue. The IP (which is hard to interpret due to the extra lane that is shown for some reason) you also IP less VPS35, so perhaps the protein is less stable? For the cargo tracking experiments, have they validated using their in vitro system that VPS10 is indeed a SNX-BAR cargo and that Ear1 and Ste13 are Snx3 cargos (I apologise if I missed this)? In addition, this data would be equally consistent with a handover mechanism and otherwise heavily relies on the VPS35 mutation, which I think unless further validated is weak.

My next concern is with the pitch of the manuscript. I am a huge supporter of yeast biology, and of course, the similarities between the yeast and metazoan internal organisation and endolysosomal and secretory systems are what have made it such a fruitful model; however, in this case, I think there are some limitations. The organisation of retromer is genuinely different in yeast and metazoa, the proteins that are conserved interact differently, and there are additional complex systems such as WASH and VARP (which is in some yeast, I think, but not the standard lab strains). I think it is really important that the findings here are not implied to be conserved in metazoa or are directly tested in them.

The in vitro reconstitutions have been performed with exemplary rigour; however, I have some concerns with some of the other

parts of the manuscripts that included the quantifications and statistics. Frequently, observations are made and interpreted without quantification; there are no statistics in comparisons that are made; cells are treated as biological repeats (where the mean of the biological repeat should be made from a number of cells); stats tests when performed are not detailed in the legend; numbers are given with no idea of the actual variability. I really think this natively affects the perception of the manuscript- if experiments were repeated three times as stated, the data should be quantified and shown.

In places, I struggled to interpret the manuscript. Some examples are: "immunoabsorption" (an unusual term) is used to describe what is described in other places as an immunoprecipitation. In that figure, there is an extra lane shown for some reason with text on it (7C). The IP is described as RFP when it is the mCherry that is IP'd, not RFP.

Finally, I thought the discussion was well written. The section on the additional challenges to sorting is key and will be well read by the field. I lightly encourage the authors to sharpen this section. The TMD explanation does not parse as in their model, the TMD cargos will be on the same carriers- in my opinion, this idea just confuses the issue and it should be removed. The simple truth is, what they are saying is quite extraordinary when it comes down to it as it only leaves two options: 1) Endosomal sorting is not that selective to location and sorting happens on other organelles (PM or Golgi) or 2) There is a second sorting step after the one detailed here. If I have missed something with my thinking, I hope to clearly read it in the revised manuscript.

Referee #2:

Gopaldass and colleagues from the University of Lausanne in the laboratory of Andreas Mayer have investigated the formation of hybrid coats of retromer with the sorting nexins SNX-BAR and Snx3 (EMBOJ-2025-121998). Using model membrane system that the team had pioneered and further explorations in yeast, it is concluded that retromer may recruit these two types of sorting nexins into the same sorting structures, possibly to enhance the membrane curvature active properties of both coats.

Endosomal sorting remains a dynamic field of study in membrane biology. The current manuscript thereby appears particularly timely. The experimental approach is highly elegant, and the in vitro system is impressive through the possibility of obtaining quantitative information on protein recruitment and curvature radii. The current study should be of high interest to a general readership in molecular and cellular biology.

The following aspects might need to be addressed to clarify two remaining points.

The authors observe that membrane coats that are preformed in vitro on SMTubes with one type of sorting nexin cannot integrate the respectively other type of sorting nexin upon incubations in the minutes time range. It should ideally be experimentally tested or at least discussed how parameters such as membrane composition (notably in vivo-like lipid mixtures), tube diameter, and membrane tension could have an influence here.

The conclusion that hybrid coat formation allows Snx3 and SNX-BARs to provide higher membrane scaffolding activity than in respective homogeneous coats (lines 288-290), does that also hold when cargos are present on the membrane?

Response to the reviewers' comments:

We thank both reviewers for their careful and overall very positive assessment of our study. Their comments led us to improve the manuscript in several aspects. We provide our responses and describe our changes below the reviewers' comments, which we have marked in bold type.

Referee #1:

Gopaldass et al present a set of data addressing one of the fundamental open questions in endosome biology- how do the litany of retrograde trafficking complexes and retromer flavours organise themselves? Using two complexes, the SNX-BAR:retromer and SNX3:retromer from yeast, they use an elegant in vitro system to make a number of important and interesting observations. The most interesting of which is that in the situation where both complexes are co-incubated on membranes, they co-localise to the same membrane patch, and moreover, there appears to be a constriction-property benefit from co-ordination of the two complexes. I find this finding extremely compelling, and it really challenges the retromer flavour 'handover' model that has become the accepted narrative in the last ten years. I have some issues with the manuscript and particularly the in vivo section, which I will detail below; however, overall, I find these experiments and the findings compelling and important.

I will start off by discussing the in vivo data, which I think is one of the weaker sections of what is an overall very well-performed and interoperated set of data and manuscript. Simply put, there is nothing presented to my mind which actually directly supports the in vitro data. I will list my criticisms of each of the supporting experiments. The colocalisation microscopy is not surprising that they colocalise somewhat; it is diffraction-limited imaging in a small model organism, regardless of the actual situation, they would be on the same endosome/endosomal subdomain. Are they looking at carriers here? Is there any evidence of this?

We agree with the reviewer to some degree. It is hard to visualize endosomal carriers forming, even in mammalian cells, and it is virtually impossible in yeast. So, yes, our images do not show carrier formation. We did not want to claim that in the manuscript and added the colocalization experiments only to demonstrate to the reader that Snx3 and SNX-BARs are present on the same organelles, so that there is potential to co-integrate into a carrier. We now make this more explicit in the text.

The other colocalization approach that we show is meant to quantify mislocalization of cargo to the vacuoles, which we use as an indicator for the functional cross-dependency of Snx3 and SNX-BARs. Here, optical resolution is not a problem. We consider this functional cross-dependency, even if it is limited, as one of the two elements by which the in vivo approach can complement the in vitro observations and connect them to the physiological situation. The other one is the co-IP approach, because it suggests that, in vivo, Retromer can bridge Snx3 and SNX-BARs, testing a physical feature that might be expected from a hybrid coat. Please note also that reviewer 2 has underlined this as a particular strength.

As an aside, there is no negative control for the Pearsons, but this is a minor issue.

It is indeed important to exclude that the assay drowned in excessive background. Therefore, have now added the same colocalization analysis in vps35 knockout cells. These cells show reduced colocalization between Snx3 and Vps5, providing a negative control for comparison.

The IP (which is hard to interpret due to the extra lane that is shown for some reason) you also IP less VPS35, so perhaps the protein is less stable?

The extra lane is indicating the MW markers, some of which stain on the blot due to their very large abundance and cross-reactivity. To avoid confusion, we have removed them now.

We assume that the reviewer refers to Fig. 7E with the comment that the IP contains less Vps35. This may be a misunderstanding. What has been pulled down in this experiment is not Vps35 but Snx3-V5. This is indicated in the figure legend. The reviewer may have overlooked it. By pulling on Snx3-V5 we seek to demonstrate that it interacts with Vps5 and that a mutation in Vps35, which destabilizes the Vps35-Snx3 interface (this effect is shown in Fig. 7D, where we do pull on Vps35) reduces this interaction. This suggests that Vps35 can bridge Snx3 and SNX-BARs.

For the cargo tracking experiments, have they validated using their in vitro system that VPS10 is indeed a SNX-BAR cargo and that Ear1 and Ste13 are Snx3 cargos (I apologise if I missed this)?

The reviewer may indeed have missed this. Figure 1A shows that Ear1 or Ste13 must be present to allow pure Snx3-Retromer (without the SNX-BARs) to form coats, which implies that they interact with this coat.

For the SNX-BARs, we had indeed not shown a separate figure showing that they are by themselves capable of concentrating Vps10. This information has now been added as new Supplementary Fig. 2.

In addition, this data would be equally consistent with a handover mechanism and otherwise heavily relies on the VPS35 mutation, which I think unless further validated is weak.

We have validated the effect of the vps35^{QGRE} allele in Fig. 7D, which shows a Vps35-HA pulldown. Co-adsorption of Snx3, but not of Vps5, is strongly reduced by vps35^{QGRE}, suggesting that this mutation selectively destabilises the interaction with Snx3. This is also supported by a study from the Burd lab, which we cite. It showed that the region of Vps35 that we targeted by point mutations can be chemically crosslinked to Snx3. We designed this mutant based on available structural information from Leneva et al. and from Hierro et al., as well as by modelling the complex with AlphaFold 3. We have now added Suppl. Fig. 3 to illustrate the predicted interaction site and the substituted residues.

Concerning the handover mechanism: Our experiments are not apt to address a potential handover. We find handover an attractive concept, which was put forward but is so far supported by limited evidence using overexpressed cargo. In our in vitro experiments, both pure Snx3 and pure SNX-BAR Retromer coats incorporate their respective cargos quite well (see new Suppl. Fig. 2), suggesting that they might not depend on a handover. However, that argument would be weak since dependence might arise under optimized conditions. For these reasons, we prefer not to open a discussion of the handover model at this point. We are, however, performing a dedicated study of cargo effects on coat formation, where this point may come up and be more appropriate.

My next concern is with the pitch of the manuscript. I am a huge supporter of yeast biology, and of course, the similarities between the yeast and metazoan internal organisation and endolysosomal and secretory systems are what have made it such a fruitful model; however, in this case, I think there are some limitations. The organisation of retromer is genuinely different in yeast and metazoa, the proteins that are conserved interact differently, and there are additional complex systems such as WASH and VARP (which is in some yeast, I think, but not the standard lab strains). I think it is really important that the findings here are not implied to be conserved in metazoa or are directly tested in them.

We agree. The mammalian system is more complex, and the impact of this complexity will have to be addressed in the future. We now finish our discussion highlighting that the metazoans have a greater variety of endosomal coat systems and that these are also more complex, indicating a need for additional studies to explore their potential for forming hybrid coats.

The in vitro reconstitutions have been performed with exemplary rigour; however, I have some concerns with some of the other parts of

the manuscripts that included the quantifications and statistics. Frequently, observations are made and interpreted without quantification; there are no statistics in comparisons that are made; cells are treated as biological repeats (where the mean of the biological repeat should be made from a number of cells); stats tests when performed are not detailed in the legend; numbers are given with no idea of the actual variability. I really think this natively affects the perception of the manuscript- if experiments were repeated three times as stated, the data should be quantified and shown.

Experiments were all repeated 3 times. For the Pearson colocalization analysis (several fields containing 20 to 30 cells) per experiments were analysed. On the graphs, a data point represents the pearson coefficient for a single field. We chose to show all individual points rather than calculating the mean per experiment. We now provide an alternative representation of the data, which the reviewer alludes to, in Suppl. Fig. 7, which gives an impression of the variation between experiments.

In places, I struggled to interpret the manuscript. Some examples are: "immunoadsorption" (an unusual term) is used to describe what is described in other places as an immunoprecipitation. In that figure, there is an extra lane shown for some reason with text on it (7C). The IP is described as RFP when it is the mCherry that is IP'd, not RFP.

Thank you. These inconsistencies have been corrected.

Finally, I thought the discussion was well written. The section on the additional challenges to sorting is key and will be well read by the field. I lightly encourage the authors to sharpen this section. The TMD explanation does not parse as in their model, the TMD cargos will be on the same carriers- in my opinion, this idea just confuses the issue and it should be removed. The simple truth is, what they are saying is quite extraordinary when it comes down to it as it only leaves two options: 1) Endosomal sorting is not that selective to location and sorting happens on other organelles (PM or Golgi) or 2) There is a second sorting step after the one detailed here. If I have missed something with my thinking, I hope to clearly read it in the revised manuscript.

We appreciate this comment, which incited us to extend and sharpen the discussion.

The TMD may become important as a targeting element not so much for determining the integration of the cargo into the carrier. We assume that it may rather define whether a cargo will be retained at a target compartment that the carrier fuses with, or whether the cargo will undergo re-location, using other trafficking events, until it finally reaches the compartment where it fits. In the revised discussion, we hope to address this and the other

points mentioned more clearly.

Referee #2:

Gopaldass and colleagues from the University of Lausanne in the laboratory of Andreas Mayer have investigated the formation of hybrid coats of retromer with the sorting nexins SNX-BAR and Snx3 (EMBOJ-2025-121998). Using model membrane system that the team had pioneered and further explorations in yeast, it is concluded that retromer may recruit these two types of sorting nexins into the same sorting structures, possibly to enhance the membrane curvature active properties of both coats.

Endosomal sorting remains a dynamic field of study in membrane biology. The current manuscript thereby appears particularly timely. The experimental approach is highly elegant, and the in vitro system is impressive through the possibility of obtaining quantitative information on protein recruitment and curvature radii. The current study should be of high interest to a general readership in molecular and cellular biology.

The following aspects might need to be addressed to clarify two remaining points.

The authors observe that membrane coats that are preformed in vitro on SMTubes with one type of sorting nexin cannot integrate the respectively other type of sorting nexin upon incubations in the minutes time range. It should ideally be experimentally tested or at least discussed how parameters such as membrane composition (notably in vivo-like lipid mixtures), tube diameter, and membrane tension could have an influence here.

We agree with this point in principle. It reflects a general limitation of all in vitro work that the conditions chosen for the experiments may influence the results. Nevertheless, we have added text to the discussion section to briefly highlight this point. We inserted only two phrases because we are reluctant to speculate about the potential impact of these and other factors in detail. Addressing them experimentally in a meaningful way would essentially require a dedicated, separate study.

The conclusion that hybrid coat formation allows Snx3 and SNX-BARs to provide higher membrane scaffolding activity than in respective homogeneous coats (lines 288-290), does that also hold when cargos are present on the membrane?

Yes, it does. All experiments that we showed with Snx3, unless stated otherwise, have been performed in the presence of Ear1 cargo peptide. To strengthen this point, we have repeated the GUV experiment in the presence of both cargoes. The effect holds in this case as well. We have included this data in the new Supplementary figure 3.

Dear Prof. Mayer,

We have now received re-review reports from two referees, which I have included below. As you will see, you have addressed their concerns satisfactorily. However, before I can finally accept the manuscript, there are some remaining editorial points which need to be addressed. In this regard would you please:

- include up to five keywords
- include a "Disclosure and competing interests statement"; employment in a biotech company needs to be mentioned here,
- refer to Fig. 6A-B and EV figure panels in the text,
- include a Reagents and Tools table,
- provide source data for Fig. 8B, 8D and 8F; source data files need to be saved in a scheme of one figure per folder and then uploaded as .zip files. E.g. all the Source data files for figure 1 need to be saved in a single folder and this needs to be zipped and then uploaded as "SD figure 1.zip" file. For EV and/or appendix figures, ZIP together all source data,
- upload the completed source data checklist as a Related Manuscript File,
- define the annotated p values ****/**/**/* and provide the exact p-values in the legend of figure 7B, D, F, 8B, D, F; EV6 B, D; EV7 A-C as appropriate,
- state the statistical test used for data analysis in the legends of figures 7D, F,
- define n in the legends of figures 1G, 3D, 5E, H; 7D, F,
- define error bars in the legends of figures 1G, 3D, 5C, E, H; 6B, 7D, F; EV3 B,
- rename Appendix Table 1-3 as Table 1-3,
- rename "Materials and Methods" as "Methods", and
- use the following section order: Title page - Abstract - Keywords - Introduction - Results - Discussion - Methods - Data Availability - Acknowledgements - Disclosure and Competing Interests Statement - References - Figure Legends - Tables - Expanded View Figure Legends.

We include a synopsis of the paper (see <http://emboj.embopress.org/>). Please also provide me with a general summary image, a two sentence statement and 3-5 bullet points that capture the key findings of the paper.

I am looking forward to receiving your revised manuscript.

EMBO Press is an editorially independent publishing platform for the development of EMBO scientific publications.

Best wishes,

William

William Teale, PhD
Editor
The EMBO Journal
w.teale@embojournal.org

Read our guidance for manuscript revisions and related editorial policies: <https://link.springer.com/journal/44318/submission-guidelines#cms-Revised-submissions>

<https://media.springernature.com/original/springer-cms/rest/v1/content/27825798/data/v1>

- a point-by-point response to the referees' comments, with a detailed description of the changes made (as a word file).
- a word file of the manuscript text.
- individual production quality figure files (one file per figure)
- a complete author checklist
- Expanded View files (replacing Supplementary Information)

- a Reagents and Tools Table as part of the Methods section

Please remember: Digital image enhancement is acceptable practice, as long as it accurately represents the original data and conforms to community standards. If a figure has been subjected to significant electronic manipulation, this must be noted in the figure legend or in the 'Methods' section. The editors reserve the right to request original versions of figures and the original images that were used to assemble the figure.

We realize that it is difficult to revise to a specific deadline. In the interest of protecting the conceptual advance provided by the work, we recommend a revision within 3 months (23rd Mar 2026). Please discuss the revision progress ahead of this time with the editor if you require more time to complete the revisions. Use the link below to submit your revision:

Referee #1:

The revised manuscript and attached response by Gopaldass et al addresses all my concerns. I congratulate the authors for an important contribution and fully support its publication which I suspect will be a fundamental and exciting contribution as we start to understand the coordination of the multiple endosomal sorting complexes.

Referee #2:

The authors have responded in full to the comments that I had on the initial version of the manuscript.

All minor editorial requests have been addressed by the authors.

Dear Andreas,

I am pleased to inform you that your manuscript has been accepted for publication in the EMBO Journal.

Congratulations to you and your team!

You may qualify for financial assistance for your publication charges - either via a Springer Nature fully open access agreement or an EMBO initiative. Check your eligibility: <https://link.springer.com/journal/44318/how-to-publish-with-us>

Best wishes,

William

William Teale, PhD
Editor
The EMBO Journal
w.teale@embojournal.org

Please note that it is The EMBO Journal policy for the transcript of the editorial process (containing referee reports and your response letters) to be published as an online supplement to each paper. If you should prefer removal of any referee-only figures included in the point-by-point response(s), e.g. because they may still be used for future publication or because they have been reproduced from published work by others, please do let us know immediately via response email.

More information is available here: <https://link.springer.com/partners/embo-press/editorial-policies#Peer%20review>